# Fast and Scalable Analytical Diffusion

**Xinyi Shang** [* 1 2]  **Peng Sun** [* 3 4]  **Jingyu Lin** [* 5]  **Zhiqiang Shen** [2]

## Abstract

Analytical diffusion models offer a mathematically transparent path to generative modeling by formulating the denoising score as an empirical-Bayes posterior mean. However, this interpretability comes at a prohibitive cost: the standard formulation necessitates a full-dataset scan at every timestep, scaling linearly with dataset size. In this work, we present the first systematic study addressing this scalability bottleneck. We challenge the prevailing assumption that the entire training data is necessary, uncovering the phenomenon of *Posterior Progressive Concentration*: the effective golden support of the denoising score is not static but shrinks asymptotically from the global manifold to a local neighborhood as the signal-to-noise ratio increases. Capitalizing on this, we propose *Dynamic Time-Aware Golden Subset Diffusion* (GOLDDIFF), a training-free framework that decouples inference complexity from dataset size. Instead of static retrieval, GOLDDIFF uses a coarse-to-fine mechanism to dynamically pinpoint the "Golden Subset" for inference. Theoretically, we derive rigorous bounds guaranteeing that our sparse approximation converges to the exact score. Empirically, GOLDDIFF achieves a **71×** speedup on AFHQ while matching or achieving even better performance than full-scan baselines. Most notably, we demonstrate the first successful scaling of analytical diffusion to ImageNet-1K [1].

## 1. Introduction

Analytical diffusion models (Kamb & Ganguli, 2024) have emerged as a principled framework for demystifying gen-

---

[*]Equal contribution [1]University College London [2]Mohamed bin Zayed University of Artificial Intelligence [3]Zhejiang University [4]Westlake University [5]Monash University. Correspondence to: Zhiqiang Shen <Zhiqiang Shen@mbzuai.ac.ae>.

*Proceedings of the $43^{rd}$ International Conference on Machine Learning*, Seoul, South Korea. PMLR 306, 2026. Copyright 2026 by the author(s).

[1]Our code is available at https://github.com/shangxinyi/GoldDiff.

erative dynamics, offering closed-form estimators that provide mechanistic insight into denoising (De Bortoli, 2022; Scarvelis et al., 2023; Kamb & Ganguli, 2024). A core primitive is the *empirical Bayes denoiser*: at each diffusion step, the model forms a posterior over the training set and uses the posterior mean as the denoising score (Kamb & Ganguli, 2024; Lukoianov et al., 2025). This transparency, however, comes with a prohibitive computational burden: the posterior is supported on all $N$ training points, so naive inference requires a full-dataset scan per timestep, yielding $\mathcal{O}(ND)$ complexity for dataset size $N$ and dimension $D$. As a result, analytical diffusion becomes impractical at scale, and unlike the extensive work on accelerating neural diffusion solvers (Sheynin et al., 2023; Niedoba et al., 2024), optimizing exact analytical estimators remains largely unexplored.

Beyond cost, scanning the entire dataset is often *unnecessary* and can even be *harmful*. In practice, only a small fraction of samples meaningfully contribute to the posterior at a given noise level (Fig. 1), while irrelevant or ambiguous points may introduce bias and statistical noise, degrading the estimate. This motivates our key idea: instead of treating all training points as equally relevant, we seek a minimal, high-quality support set that preserves the empirical Bayes score while avoiding spurious influence. We formalize this as a *golden subset*: a dynamically selected subset of training examples from a theoretical perspective that concentrates the posterior mass on the current noisy input, enabling both efficient inference and more reliable estimation.

In this work, we present the first systematic study addressing the acceleration of analytical diffusion models. We challenge the conventional assumption in analytical diffusion that accurate posterior estimation requires the entire training corpus. By systematically analyzing the spatio-temporal dynamics of posterior weights, we characterize a fundamental phenomenon termed *Posterior Progressive Concentration*, where the *golden support* (the subset of samples contributing non-negligible probability mass) evolves dramatically with the signal-to-noise ratio (Fig. 1). Specifically, in the high-noise regime, the posterior is diffuse and requires a broad support to capture global manifold structure, meaning a small static $k$ introduces severe bias (further demonstrated in Sec. 3.4). Conversely, in the low-noise regime, the posterior collapses into a tight neighborhood, rendering a large static $k$ computationally wasteful without yielding accu-

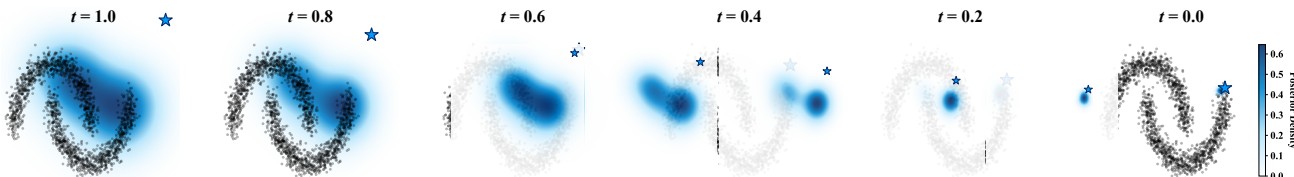

*Figure 1.* **The Phenomenon of Posterior Progressive Concentration.** ★ denotes the initial noise, and the target distribution is the Moons data (Pedregosa et al., 2011). As the diffusion process reverses from pure noise to data (Left to Right), the golden support of the posterior distribution ***dynamically*** shrinks from the global manifold to a localized neighborhood.

racy gains. This progressive concentration motivates us to decouple computational complexity from dataset size by dynamically adapting the golden support set, thereby accelerating the process without sacrificing estimation accuracy.

Building on this insight, we propose *Dynamic Time-Aware Golden Subset Diffusion* (GoldDiff). Departing from rigid static $k$-NN selection, GoldDiff employs a coarse-to-fine mechanism that dynamically pinpoints a minimal yet sufficient golden set required for accurate posterior estimation at the whole denoise time. Crucially, our method offers benefits beyond mere speed. We identify that existing state-of-the-art analytical models suffer from severe smoothing bias stemming from biased weight estimation, producing blurry outputs even with sufficient denoising steps (Fig. 2). By selectively the *Golden Subset*, we demonstrate that a simple, unbiased estimator is sufficient to unlock superior quality. Most notably, our efficiency gains unlock a milestone for the field: we successfully scale analytical diffusion to ImageNet-1K for the first time. GoldDiff achieves strong performance with a substantial acceleration, effectively bridging the gap between closed-form mathematical transparency and high-fidelity, industrial-scale generation. *Our key contributions are threefold:*

- We provide the first systematic investigation into the acceleration of analytical denoisers. We identify the phenomenon of *posterior progressive concentration*, proving that the golden support of the denoising score shrinks asymptotically as the noise level decreases. This finding has profound implications not only for analytical inference but also for the design of efficient neural denoisers.

- We propose GoldDiff, a training-free framework that dynamically retrieves a "Golden Subset" for inference. We derive rigorous theoretical bounds guaranteeing that our sparse approximation converges to the exact analytical score with controlled error.

- Extensive experiments demonstrate that the proposed GoldDiff achieves a **71× speedup** on AFHQ while matching or exceeding full-scan performance. Furthermore, we achieve the first successful scaling of analytical diffusion to large-scale ImageNet-1K, unlocking a new paradigm for scalable, interpretable generative modeling.

## 2. Related Work

**Denoising Diffusion Models.** Diffusion probabilistic models (Ho et al., 2020; Song et al., 2020b;a) have emerged as the dominant paradigm in generative modeling, achieving state-of-the-art sample quality on a wide range of benchmarks (Dhariwal & Nichol, 2021). Fundamentally, these models reverse a stochastic process that degrades the data distribution into noise. These models typically parameterize the score function $\nabla_{\mathbf{x}} \log p_t(\mathbf{x})$ using deep neural networks, commonly with U-Net backbones (Karras et al., 2022; 2024) or Transformer-based architectures (Peebles & Xie, 2023; Yao et al., 2025), trained via denoising score matching. While highly effective, such learned denoisers are often treated as "black boxes", making it difficult to interpret or mechanistically analyze the generation dynamics (Kamb & Ganguli, 2024).

**Analytical Diffusion.** Analytical Diffusion models have emerged as a rigorous framework for interpreting the generative mechanisms of diffusion models (Kamb & Ganguli, 2024). By formulating the score function as an empirical Bayes estimator, these methods enable exact score computation without black-box optimization. Seminal works (De Bortoli, 2022; Scarvelis et al., 2023) establish the optimal denoiser as a theoretical proxy for understanding the behavior of deep diffusion models. However, a central paradox lies in their generalization capabilities: the exact optimal denoiser tends to inherently memorize, collapsing to a mixture of delta functions at the training data points in the low-noise limit (Kamb & Ganguli, 2024). To explain the generalization observed in neural approximations, Kamb & Ganguli (2024) and Niedoba et al. (2025) attribute it to the inductive bias (locality) of the neural architectures. Conversely, Lukoianov et al. (2025) recently posit that such locality emerges as an intrinsic statistical property of the training dataset. Leveraging this insight, they incorporate their analytically-computed locality into the optimal denoiser-based model, demonstrating superior performance over standard analytical baselines.

**Efficient Inference and Retrieval-Augmented Diffusion.** Accelerating diffusion inference is a longstanding challenge. While standard techniques focus on reducing the number of timesteps (e.g., distillation (Salimans & Ho, 2022; Yin et al.,

2024a;b)), a parallel line of research aims to reduce the per-step cost by approximating the data distribution via coresets or varying clusters (Niedoba et al., 2024). In the context of retrieval-based methods, static $k$-Nearest Neighbors ($k$-NN) serves as a prevalent approximation strategy (Niedoba et al., 2024; Sheynin et al., 2023). For example, KNN-Diffusion (Sheynin et al., 2023) and RetrievalAugmented Diffusion Models (Blattmann et al., 2022) both condition diffusion models on the fixed $k$ nearest CLIP (Radford et al., 2021) embeddings of clean training images. Our work differs in *two* fundamental aspects. First, we pioneer the optimization of analytical estimators. Second, we challenge the static retrieval paradigm and offer a theory-grounded time-varying posterior estimation framework.

## 3. Time-Aware Golden Subset Diffusion

In this section, we present GOLDDIFF, a training-free acceleration framework for analytical diffusion. To effectively retrieve the "Golden Subset," we first revisit the analytical denoisers (Sec. 3.1) and explore the impact of biased weight estimation (Sec. 3.2). Then, we analyze the sensitivity to subset selection, revealing a distinct *two-regime behavior* governed by the signal-to-noise ratio (Sec. 3.3). Motivated by these insights, we propose a theoretical-grounded dynamic selection strategy that optimizes the trade-off between retrieval recall and the aggregation budget via time-aware schedules (Sec. 3.4). Furthermore, we establish rigorous theoretical guarantees to validate our dynamic mechanism and provide complexity analysis (Sec. 3.5).

### 3.1. Revisiting Analytical Denoisers

We define the forward diffusion process as $\mathbf{x}_t = \sqrt{\alpha_t}\mathbf{x}_0 + \sqrt{1-\alpha_t}\epsilon$, where $\alpha_t$ follows a monotonic signal schedule and $\epsilon \sim \mathcal{N}(\mathbf{0}, \mathbf{I})$. The standard MSE training objective is minimized by the conditional expectation of the clean data (Vincent, 2011; Ho et al., 2020):

$$\min_{\mathbf{f}} \mathbb{E}\left[\|\mathbf{f}(\mathbf{x}_t, t) - \mathbf{x}_0\|^2\right] \implies \hat{\mathbf{f}}(\mathbf{x}, t) = \mathbb{E}[\mathbf{x}_0 \mid \mathbf{x}_t = \mathbf{x}]. \quad (1)$$

**Optimal Empirical Bayes Denoising.** By modeling the data prior as the empirical distribution of the training set $\mathcal{D} = \{\mathbf{x}_i\}_{i=1}^N$, i.e., $p(\mathbf{x}_0) = \frac{1}{N}\sum_i \delta(\mathbf{x}_0 - \mathbf{x}_i)$, the exact *Empirical Bayes Denoiser* admits a closed-form solution as a kernel-weighted average (Kamb & Ganguli, 2024; De Bortoli, 2022; Lukoianov et al., 2025) :

$$\hat{\mathbf{f}}(\mathbf{x}, t) = \sum_{i=1}^N \underbrace{\text{softmax}_i \left(-\frac{\|\mathbf{x}/\sqrt{\alpha_t} - \mathbf{x}_i\|^2}{2\sigma_t^2}\right)}_{\mathbf{w}_i(\mathbf{x},t)} \cdot \mathbf{x}_i, \quad (2)$$

where $\sigma_t^2 := (1 - \alpha_t)/\alpha_t$ is the noise-to-signal ratio. Eq. (2) interprets the optimal denoiser as a *distance-weighted average over the global training set*.

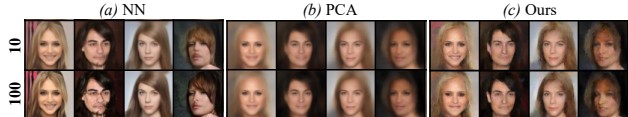

*Figure 2.* **Impact of Biased Weight Estimation.** Due to biased weight estimation, PCA (Lukoianov et al., 2025) produces inherently over-smoothed outputs even with sufficient denoising steps.

**Manifold Locality and Generalization.** While Eq. (2) is optimal in the MMSE sense, Kamb & Ganguli (2024) reveals that it suffers from severe memorization, tending to reproduce training samples rather than generating novel content. They reveal that the generalization ability of diffusion models arises from *locality*, proposing a patch-based approximation that restricts attention to local spatial neighborhoods. Building on this, Lukoianov et al. (2025) further demonstrates that locality is an intrinsic statistical property of the image manifold. They propose projecting data onto local Principal Component Analysis (PCA) bases to better capture this manifold structure. Generalizing these local variants, the denoising score can be expressed as:

$$\hat{\mathbf{f}}(\mathbf{x}_t, t) = \sum_{i=1}^N \tilde{\mathbf{w}}_i(\mathbf{x}_t) \cdot \mathcal{P}_i(\mathbf{x}_i), \quad (3)$$

where $\mathcal{P}_i$ denotes a generalized local operator (e.g., patch extraction in (Kamb & Ganguli, 2024) or PCA projection in (Lukoianov et al., 2025)), and $\tilde{\mathbf{w}}_i$ represents the weights computed within that local subspace. Crucially, regardless of the specific projection $\mathcal{P}_i$, the computational bottleneck remains the summation over $N$ training samples, which scales linearly as $\mathcal{O}(ND)$, rendering exact inference intractable for large-scale datasets.

### 3.2. Unbiased Weight Estimation

The current state-of-the-art PCA method (Lukoianov et al., 2025) computes posterior weights over the *entire* training set. In practice, the dataset is highly heterogeneous: some samples are truly informative, others are largely irrelevant, and a nontrivial portion can even be harmful (e.g., noisy or misleading neighbors). This global weighting often produces an overly sharp, heavy-tailed weight distribution dominated by a few points, especially at the late denoising stage (visualized in Fig. 1). To avoid such sharp weights and the resulting numerical instability, PCA adopts a biased weighted streaming softmax with batch-level averaging to manually flatten the weights. However, we find that it introduces a systematic smoothing bias that weakens the analytical denoiser and reduces fidelity. As qualitatively evidenced in Fig. 2 (b) and the fourth row of Fig. 4, PCA struggles to recover high-frequency details and tends to produce blurry outputs even with sufficient denoising steps.

Our approach is fundamentally different and more elegant: instead of correcting an ill-conditioned global weighting with a biased softmax, we first fix the support. Using

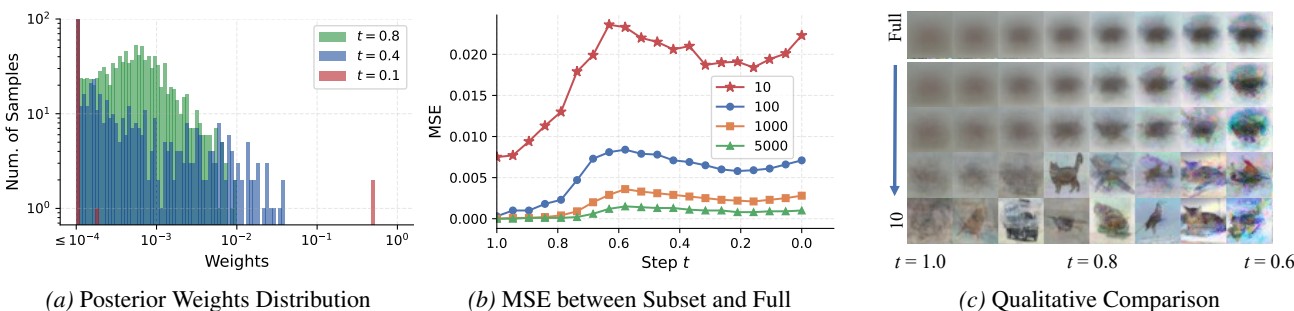

*(a)* Posterior Weights Distribution      *(b)* MSE between Subset and Full      *(c)* Qualitative Comparison

*Figure 3.* **Analysis of the SOTA method PCA (Lukoianov et al., 2025) for Subset Selection.** (a) Evolution of the weight distribution during the denoising process. (b)–(c) Sensitivity analysis: (b) Performance evaluation of the analytical denoiser across varying random subset sizes ($N_{\text{sub}} \in \{10, 100, 1000, 5000\}$) compared to the full CIFAR-10 dataset. (c) Visualization of intermediate generation outputs.

GOLDDIFF, we dynamically filter a *Golden Subset*, a compact set of samples that are useful, information-rich, and reliable for the current noisy input. Once the support is purified in this way, we no longer need manual weight-flattening tricks: we can directly apply a simple and unbiased streaming softmax (Dao et al., 2022) on the *Golden Subset*. This naturally regulates the weight distribution, decoupling stability from bias, and yields sharper details and better semantic coherence without the over-smoothing effects of weighted softmax or the instability risks of naive full-corpus weighting. As qualitatively demonstrated in Fig. 2(c) and the fifth row of Fig. 4, the samples generated via GOLDDIFF achieve significantly better alignment with neural denoisers than previous analytical baselines. A further ablation study and analysis are provided in Sec. 4.3.

### 3.3. Why Dynamic Retrieval? A Sensitivity Analysis

To determine the optimal allocation of computational budgets, we investigate the mechanics of the estimator across the diffusion steps. Combining the analysis of weights evolution in Fig. 3a and the denoiser's sensitivity to subset scaling in Fig. 3b, we characterize two distinct regimes governed by the Signal-to-Noise Ratio.

**Early Stage: Global Manifold Approximation.** In the early stages of diffusion, the signal is drowned out by noise ($\sigma_t^2 \gg 1$). Fig. 3a shows a diffuse posterior where no single sample dominates. Crucially, our sensitivity analysis in Fig. 3b reveals that performance drops significantly with small subsets ($N = 10, 100$) but recovers with a larger random subset ($N = 1000$). Qualitatively, as shown in Fig. 3c, restricting the support to a small subset (e.g., $N = 10$) during this phase yields reconstructions that manifest as blurred superpositions of the training samples, failing to capture the global data manifold. Conversely, utilizing a larger subset produces outputs indistinguishable from those derived using the full dataset. These observations imply that in this regime, the estimator functions as a Monte Carlo Integrator: it relies on the *Law of Large Numbers* to approximate the global expectation $\mathbb{E}[\mathbf{x}_0]$. Therefore, the estimator is robust to *retrieval imprecision* (random selection works if $N$ is

large) but sensitive to *sample sparsity*. Thus, we need a *broad support* ($k \uparrow$) to cover the manifold in this regime, but can rely on *coarse retrieval*.

**Late Stage: Local Neighbor Selection.** As the noise vanishes ($\sigma_t^2 \to 0$), Fig. 3a demonstrates a sharp entropy collapse, where probability mass concentrates on a tiny neighborhood. Here, the estimator shifts from *integration* to *selection*. The Weight Gap between the true neighbor and distant samples explodes, meaning omitting the Top-1 neighbor introduces catastrophic bias. The estimator becomes robust to *sample sparsity* (a few samples suffice) but extremely sensitive to *retrieval precision*. Thus, we need *high-precision retrieval* ($m \uparrow$) but can enforce *aggressive sparsity* ($k \downarrow$). Driven by this "Integration-to-Selection" transition, GOLDDIFF employs a time-aware mechanism that dynamically allocates the computational budget. We propose a *Counter-Monotonic Schedule* for retrieval scope ($m_t$) and aggregation budget ($k_t$).

### 3.4. Theoretical-grounded Dynamic Selection

**Adaptive Coarse Screening.** We first filter the full dataset $\mathcal{D}$ to a candidate set $\mathcal{C}_t$ of size $m_t$ using a computationally efficient proxy metric. Specifically, we employ a spatially downsampled $\ell_2$ distance: $d^{\text{proxy}}(\mathbf{x}_t, \mathbf{x}_i) \triangleq \|\operatorname{Down}_s(\mathbf{x}_t) - \operatorname{Down}_s(\mathbf{x}_i)\|_2$, where $s = 1/4$. This design leverages the *hierarchical consistency* of natural images (Wang et al., 2004): distinct local similarity typically correlates with reasonable proximity in the low-frequency structure captured by our proxy. Furthermore, to guarantee recall in the low-noise regime (where exact matches are critical), the candidate pool size $m_t$ must expand as noise decreases. We model $m_t$ via a monotonically *increasing* schedule with respect to signal strength:

$$m_t = \lfloor m_{\min} + (m_{\max} - m_{\min}) \cdot (1 - g(\sigma_t)) \rfloor, \quad (4)$$

where $g(\sigma_t) \in [0, 1]$ is the normalized noise level. This ensures $m_t \to m_{\max}$ as $t \to 0$, providing a "safety margin" to recall true neighbors when precision is paramount.

**Precision Golden Set Selection .** We then compute exact

*Table 1.* **Algorithmic Complexity Comparison.** $D$: flattened image dimension, $N$: dataset size, $p_t$: patch size at step $t$.

| Method | Optimal | Wiener Filter | Kamb | PCA | **GOLDDIFF (Ours)** |
|---|---|---|---|---|---|
| Complexity | $\mathcal{O}(ND)$ | $\mathcal{O}(D^2)$ | $\mathcal{O}(Np_tD^2)$ | $\mathcal{O}(Np_tD)$ | $\mathcal{O}(\mathbf{Nd} + \mathbf{m_t p_t D})$ |

distances strictly within $\mathcal{C}_t$ to obtain the final golden support $S_t$ of size $k_t$:

$$S_t = arg\,topk_{i\in\mathcal{C}_t, |S|=k_t}\left\{-\|\mathbf{x}_t - \mathbf{x}_i\|_2\right\}. \quad (5)$$

Aligning with the posterior concentration (Fig. 1), the aggregation budget $k_t$ follows an opposing schedule:

$$k_t = \lfloor k_{\min} + (k_{\max} - k_{\min}) \cdot g(\sigma_t) \rfloor. \quad (6)$$

As $t \to 0$, $k_t \to k_{\min}$, exploiting the sparsity of the posterior. This symmetric trade-off minimizes complexity, achieving significant acceleration. Our analysis in Sec. 4.3 demonstrates that the optimal values for these hyperparameters are empirically consistent across multiple datasets.

### 3.5. Truncation Error Bound and Complexity Analysis

We rigorously bound the approximation error induced by truncating the support to a subset $S_t$. Let $\hat{\mathbf{f}}_{\mathcal{D}}(\mathbf{x}_t)$ and $\hat{\mathbf{f}}_{S_t}(\mathbf{x}_t)$ denote the exact and truncated posterior means, respectively. We derive the following upper bound (see App. A for the detailed derivation):

> **Theorem 1 (Posterior Truncation Error Bound).** *Assume logits $\ell_i$ are sorted such that $\ell_{(1)} \geq \cdots \geq \ell_{(N)}$. If the estimator aggregates only the top-k samples, the error is bounded by:*
>
> $$\|\hat{\mathbf{f}}_{\mathcal{D}}(\mathbf{x}_t) - \hat{\mathbf{f}}_{S_t}(\mathbf{x}_t)\|_2 \leq 2R(N-k)\cdot\exp\left(-\Delta_k\right), \quad (7)$$
>
> *where $R = \max_{\mathbf{x}\in\mathcal{D}}\|\mathbf{x}\|_2$ is the data radius, and $\Delta_k \triangleq \ell_{(1)} - \ell_{(k+1)}$ represents the Logit Gap.*

The bound reveals why our dynamic $k_t$ schedule is theoretically sound: In the high-noise regime ($t \to T$), the noise term in the denominator of $\ell_i$ is large, suppressing differences between samples. Consequently, the Logit Gap $\Delta_k \to 0$, and the exponential term $\exp(-\Delta_k) \to 1$. The error bound becomes linear in $(N-k)$. To control the error, we must maximize $k$ (i.e., $k \to k_{\max}$), as prescribed by Eq. (6). Conversely, in the low-noise regime ($t \to 0$), the Logit Gap widens significantly ($\Delta_k \gg 1$). The error term $\exp(-\Delta_k)$ decays exponentially, rendering the tail contribution negligible. This allows for aggressive truncation strategy ($k \to k_{\min}$), ensuring that computational efficiency is gained without performance loss.

We summarize the algorithmic complexity in Tab. 1. Standard methods such as Kamb (Kamb & Ganguli, 2024) and PCA (Lukoianov et al., 2025) typically scale linearly with the dataset size $N$ for every pixel or patch (e.g., $\mathcal{O}(Np_tD)$),

rendering them computationally prohibitive for large-scale data. By introducing a low-dimensional proxy space $\mathbb{R}^d$ (where $d \ll D$) and a dynamic search scope $m_t \ll N$, GOLDDIFF reduces the complexity. Crucially, GOLDDIFF functions as a plug-and-play module and can be seamlessly integrated into existing baselines (detailed in Sec. 4.2). For instance, when applied to the PCA(Lukoianov et al., 2025), it effectively reduces the complexity to $\mathcal{O}(Nd + m_t p_t D)$.

## 4. Experiments

Our evaluation is guided by three core questions: (1) *Efficacy*: Does the dynamic "Golden Subset" approximation converge to the exact analytical score derived from the full dataset? (2) *Efficiency*: Can GOLDDIFF significantly reduce the computational complexity? (3) *Scalability*: Can GOLDDIFF effectively scale to large-scale benchmarks?

### 4.1. Experimental Setup

**Datasets.** We employ a comprehensive evaluation protocol spanning diverse resolutions and complexities. Benchmarks include standard grayscale datasets (MNIST (Deng, 2012), FashionMNIST (Xiao et al., 2017); $28 \times 28$), natural images (CIFAR-10 (Krizhevsky et al., 2009); $32 \times 32$), and high-resolution structured domains (CelebA-HQ (Karras et al., 2017), AFHQv2 (Choi et al., 2020); $64 \times 64$). Crucially, we extend analytical denoising evaluation to the large-scale ImageNet-1K dataset (Deng et al., 2009) ($64 \times 64$) for the first time, reporting results of both unconditional and class-conditional generation.

**Baselines.** We compare GOLDDIFF against a hierarchy of analytical and neural denoisers:

- *Analytical Denoisers:* We benchmark against the classical Wiener filter (Wiener, 1949), the Optimal Denoiser (De Bortoli, 2022), the patch-based method by Kamb et al.[2] (Kamb & Ganguli, 2024), and the state-of-the-art PCA denoiser (Lukoianov et al., 2025).

- *Neural Denoisers:* To assess the efficacy of analytical denoisers, following (Kamb & Ganguli, 2024; Lukoianov et al., 2025), we utilize the DDPM U-Net (Ho et al., 2020) (with self-attention removed for fair architectural comparison) as the ground truth oracle. Furthermore, we extend the evaluation protocol by additionally employing the strong EDM (Karras et al., 2022).

**Evaluation Metrics.** We assess performance along two axes: *(1) Efficacy:* Following previous works (Kamb & Ganguli, 2024; Lukoianov et al., 2025), we report Mean Squared Error (MSE) and the coefficient of determination ($r^2$) to quantify alignment between the analytical score estimate and the neural oracle. Higher $r^2$ ($\uparrow$) and lower MSE

---

[2]We refer to this method as "Kamb" for brevity, as the original paper does not propose a specific method name.

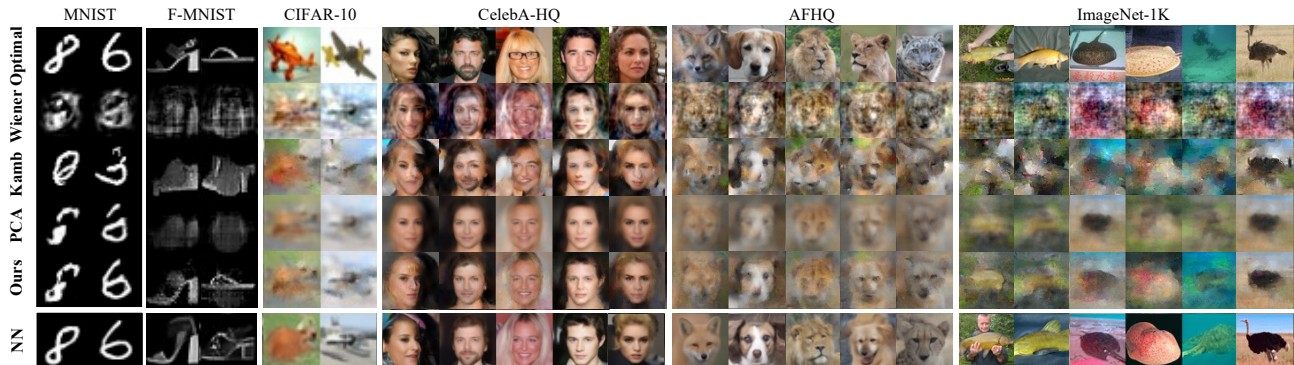

*Figure 4.* **Qualitative Comparison.** We compare our GOLDDIFF (5th row) against four baselines: Optimal (1st row), Wiener filter (2nd row), Kamb (Kamb & Ganguli, 2024) (3rd row), and the PCA model (4th row). All images are generated from the same initial noise using 10 steps of DDIM (Song et al., 2020a). The last row displays reference samples generated by a trained U-Net (Ho et al., 2020).

*Table 2.* **Quantitative Comparison of Analytical Denoisers.** Here, *Time* denotes the sampling time cost (s) per step, and *Memory* denotes the peak memory (GB) usage during sampling. All metrics are averaged over 128 samples. The best and second-best results are highlighted in **bold** and underlined, respectively. Efficacy gain (%) denotes the improvement relative to the second-best performance.

| Dataset | CIFAR-10 | | | | CelebA-HQ | | | | AFHQ | | | |
|---|---|---|---|---|---|---|---|---|---|---|---|---|
| Metric | Efficacy | | Efficiency | | Efficacy | | Efficiency | | Efficacy | | Efficiency | |
| Method | MSE ($\downarrow$) | $r^2$ ($\uparrow$) | Time | Memory | MSE ($\downarrow$) | $r^2$ ($\uparrow$) | Time | Memory | MSE ($\downarrow$) | $r^2$ ($\uparrow$) | Time | Memory |
| Optimal (De Bortoli, 2022) | 0.030 | -0.401 | 2.353 | 0.020 | 0.021 | 0.010 | 1.244 | 0.055 | 0.039 | -0.713 | 0.642 | 0.055 |
| Wiener (Wiener, 1949) | 0.008 | 0.553 | 0.011 | 0.290 | 0.012 | 0.781 | 0.153 | 4.509 | 0.010 | 0.687 | 0.156 | 4.509 |
| Kamb (Kamb & Ganguli, 2024) | 0.011 | 0.411 | 5.980 | 1.889 | 0.011 | 0.720 | 37.158 | 7.547 | 0.034 | 0.576 | 15.094 | 7.5471 |
| PCA (Lukoianov et al., 2025) | 0.008 | 0.670 | 2.802 | 0.835 | 0.009 | 0.802 | 6.040 | 5.460 | 0.008 | 0.703 | 24.896 | 4.743 |
| GOLDDIFF (Ours) | **0.007** | **0.683** | 0.087 | 0.882 | **0.008** | **0.836** | 0.349 | 5.545 | **0.007** | **0.731** | 0.351 | 4.783 |
| *vs. PCA* | $\uparrow$12.5% | $\uparrow$1.9% | $\times$**28.1** | - | $\uparrow$11.1% | $\uparrow$4.2% | $\times$**17.4** | - | $\uparrow$12.5% | $\uparrow$3.9% | $\times$**71.0** | - |

($\downarrow$) indicate better efficacy. *(2) Efficiency:* We measure time (s) per denoising step and peak memory usage (GB).

**Implementation Details.** GOLDDIFF operates as a plug-and-play module that can be seamlessly integrated into existing analytical diffusion denoisers. To demonstrate this versatility while ensuring a rigorous comparison with the state-of-the-art, we primarily report results deploying GOLDDIFF atop the PCA denoiser (Lukoianov et al., 2025). Additional results demonstrating its universality to other baselines are detailed in Sec. 4.2. Regarding the subset selection, the coarse set is dynamically expanded from $m_{min}$ to $m_{max}$, and the "Golden Subset" is reduced from $k_{max}$ to $k_{min}$ as the denoising process. In practice, for $m_{min}$ and $k_{max}$, a large random subset (e.g., 5,000 samples for CIFAR-10) achieves performance parity with the full dataset (Fig. 3b), thus we set $m_{min} = k_{max} = N/10$ where $N$ denotes the size of training set. We set $m_{max} = N/4$ and $k_{min} = N/20$ for all datasets. A detailed sensitivity analysis (Sec. 4.3) confirms that these hyperparameters are empirically consistent across multiple datasets. Following (Lukoianov et al., 2025), the number of diffusion steps is set to 10 by default.

### 4.2. Efficacy and Efficiency Comparison

We evaluate the performance of GOLDDIFF against established analytical baselines across multiple datasets, in-

cluding the small-scale datasets and the large-scale dataset ImageNet-1K (Deng et al., 2009).

**Results on Small-scale Datasets.** Quantitative metrics and qualitative comparisons are provided in Tab. 2 and Fig. 4, respectively, with additional results for MNIST and Fashion-MNIST detailed in App. B.

*Efficacy.* As evidenced in Tab. 2, GOLDDIFF consistently surpasses the state-of-the-art PCA method (Lukoianov et al., 2025) across all MSE and $r^2$ metrics. This suggests that our dynamic subset selection effectively filters out irrelevant samples, yielding more reliable score estimates. Qualitatively, Fig. 4 corroborates these quantitative gains: compared to competing baselines, GOLDDIFF exhibits superior similarity to the ground-truth neural diffusion outputs (see last row). Notably, the optimal denoiser (1st row) produces visually clear outputs. However, this stems from merely "memorizing" (Kamb & Ganguli, 2024) the training samples and fails to generalize, leading to significantly inferior MSE and $r^2$ metrics compared to our method.

*Efficiency.* Our GOLDDIFF significantly accelerates inference, achieving speedup factors of **17** $\times$ and **71** $\times$ on the CelebA-HQ and AFHQ datasets, respectively, compared to the SOTA PCA method (Lukoianov et al., 2025), while maintaining a comparable memory cost. In contrast, existing baselines struggle to balance inference speed and quality.

*Table 3.* **Quantitative Comparison on ImageNet-1K.** We evaluate performance in both unconditional and conditional settings. All metrics are averaged over 128 samples, with the best results highlighted in **bold**. "Total Steps" denotes the total number of denoising steps, and we report results for 10 and 100 steps.

| Total Steps | Method | Unconditional | | | Conditional | | |
|---|---|---|---|---|---|---|---|
| | | MSE ($\downarrow$) | $r^2$ ($\uparrow$) | Time | MSE ($\downarrow$) | $r^2$ ($\uparrow$) | Time |
| 10 | PCA | 0.045 | 0.412 | 110.798 | 0.033 | 0.467 | 0.451 |
| | PCA (Unbiased) | 0.042 | 0.435 | 110.986 | 0.032 | 0.473 | 0.453 |
| | GOLDDIFF | **0.039** | **0.458** | **2.607** | **0.031** | **0.490** | **0.100** |
| 100 | PCA | 0.032 | 0.436 | 110.798 | 0.025 | 0.521 | 0.451 |
| | PCA (Unbiased) | 0.036 | 0.425 | 110.986 | 0.035 | 0.411 | 0.453 |
| | GOLDDIFF | **0.027** | **0.509** | **2.607** | **0.022** | **0.578** | **0.100** |

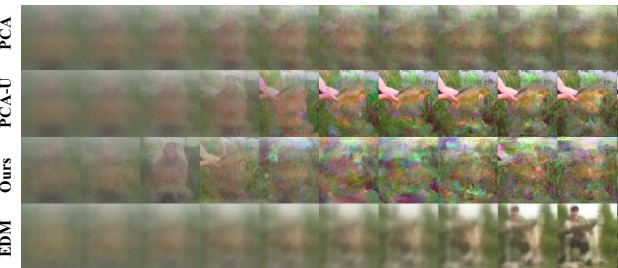

*Figure 5.* **Qualitative Comparison of conditional Denoising on ImageNet-1K.** We visualize samples generated by our method compared to two PCA-based baselines: Original PCA (Lukoianov et al., 2025) and Unbiased PCA (PCA-U) for the class "Tench".

*Table 4.* **Validation on Diverse Neural Denoisers.** All metrics are averaged over 128 samples.

| | | CIFAR-10 | | AFHQ | |
|---|---|---|---|---|---|
| | | MSE ($\downarrow$) | $r^2$ ($\uparrow$) | MSE ($\downarrow$) | $r^2$ ($\uparrow$) |
| EDM-VP | Optimal | 0.060 | -0.238 | 0.053 | 0.028 |
| | Wiener | 0.026 | 0.472 | 0.033 | 0.448 |
| | Kamb | 0.031 | 0.350 | 0.034 | 0.461 |
| | PCA | 0.023 | 0.594 | 0.029 | 0.524 |
| | GOLDDIFF | **0.0195** | **0.654** | **0.024** | **0.588** |
| EDM-VE | Optimal | 0.057 | -0.226 | 0.051 | 0.034 |
| | Wiener | 0.026 | 0.461 | 0.031 | 0.512 |
| | Kamb | 0.029 | 0.392 | 0.035 | 0.459 |
| | PCA | 0.022 | 0.604 | 0.029 | 0.535 |
| | GOLDDIFF | **0.018** | **0.666** | **0.023** | **0.595** |

While the Wiener filter is computationally efficient because its complexity depends solely on image dimensions rather than dataset size (see Tab. 1), it suffers from markedly inferior performance. On the other hand, Kamb et al. (Kamb & Ganguli, 2024) incur prohibitive computational and memory costs. By assuming translation equivariance, this method necessitates comparing the patch surrounding each pixel (of size $p_t$) against every patch in the training dataset, leading to substantial memory overhead. Furthermore, it relies on a heuristic dependency that requires estimating the effective receptive field of a pre-trained U-Net to determine the patch size $p_t$ for each diffusion timestep, which introduces considerable additional computational burden.

**Results on Large-scale ImageNet-1K.** We evaluate GOLDDIFF on large-scale ImageNet-1K under both unconditional and conditional settings, employing EDM (Karras et al., 2022) as the neural denoiser. Building upon the quantitative superiority reported in Tab. 2, we benchmark our method against the SOTA PCA (Lukoianov et al., 2025) and its unbiased variant, *PCA (Unbiased)*, which uses streaming softmax (Dao et al., 2022) for weight estimation (detailed in Sec. 3.2 and further analysis provided in Sec. 4.3).

For *Unconditional Generation*, as reported in Tab. 3, GOLDDIFF consistently achieves state-of-the-art performance across varying sampling budgets (total denoising steps $T = 10$ and $T = 100$). Moreover, in terms of efficiency, our method achieves an approximate $42\times$ acceleration in inference compared to baselines. Note that we report the inference time (s) per denoising step; thus, this per-step cost remains constant regardless of the total budget $T$.

For *Conditional Generation,* Tab. 3 provides the detailed quantitative results, where we report the mean performance and inference time averaged across all classes. Obviously, our method achieves superior performance combined with significant inference efficiency. Furthermore, we visualize the intermediate denoised images for the class "Tench" in Fig. 5. We reveal that both PCA-based baselines using the full dataset suffer from distinct failure modes rooted in their weight aggregation heuristics: (1) vanilla PCA uses the weight-averaging strategy, which leads to an *over-smoothed*

distribution and then causes the denoiser to suppress high-frequency details. This observation is consistent with the unconditional generation trends on other datasets in Fig. 2. (2) PCA (Unbiased) introduces a pathological bias toward specific training exemplars. This manifests as explicit data memorization rather than genuine manifold learning: as observed in the second row, the outputs appear as disjointed "patch-collages," often marred by anatomical incoherence (e.g., the unnatural intrusion of human hands from the training set). Crucially, increasing the denoising steps $T$ exacerbates this patch-pasting behavior, causing visual artifacts to stabilize rather than resolve. The conclusion is also verified by the results in Tab. 3, where both MSE and $r^2$ metrics decrease from $T = 10$ to $T = 100$ for conditional generation.

In contrast, our GOLDDIFF is fundamentally different and more elegant: we dynamically filter a *Golden Subset*, a compact set of samples that are useful, information-rich, and reliable for the current noisy input. Restricted to this optimized support set, this weight distribution can strike a critical balance: it circumvents the catastrophic smoothing of vanilla PCA while mitigating the memorization traps in the unbiased variant for scanning the full dataset. Consequently, as shown in Fig. 4, our method yields images characterized by sharper details and better semantic coherence without the over-smoothing effects of weighted streaming softmax or the instability risks of naive full-corpus weighting. Results reported in Tab. 3 also demonstrate its superiority.

*Table 5.* **Orthogonality to Existing Analytical Denoisers.** All metrics are averaged over 128 samples.

| Method | Celeba-HQ | | | AFHQ | | |
|---|---|---|---|---|---|---|
| | MSE ($\downarrow$) | $r^2$ ($\uparrow$) | Time | MSE ($\downarrow$) | $r^2$ ($\uparrow$) | Time |
| Optimal | 0.021 | 0.010 | 1.244 | 0.039 | -0.713 | 0.642 |
| + GOLDDIFF | **0.014** | **0.134** | **0.338** | **0.035** | **-0.270** | **0.153** |
| Kamb | 0.011 | 0.720 | 37.158 | 0.034 | 0.576 | 15.094 |
| + GOLDDIFF | **0.009** | **0.729** | **1.336** | **0.031** | **0.586** | **1.434** |

**Validation on Diverse Network Denoisers.** We assess the generality of our method GOLDDIFF by comparing it against the outputs of the state-of-the-art network-based denoisers, including EDM-VP and EDM-VE (Karras et al., 2022). As illustrated in Tab. 4, our method GOLDDIFF consistently outperforms the PCA baseline in matching the generation quality of these networks. This confirms that GOLDDIFF provides a more robust and high-quality analytical model that better aligns with the complex manifolds learned by deep neural networks.

**Orthogonality to Existing Analytical Denoisers.** We further investigate the compatibility of GOLDDIFF by incorporating it into other representative analytical baselines, including the Optimal Denoiser (De Bortoli, 2022) and Kamb (Kamb & Ganguli, 2024). Note that the Wiener filter (Wiener, 1949) is excluded from this analysis, as it relies solely on pre-computed statistics and does not require accessing the explicit training corpus during sampling. Results in Tab. 5 demonstrate that equipping these methods with GOLDDIFF not only yields superior performance but also achieves significant acceleration. This confirms that our approach acts as a versatile plug-and-play enhancement orthogonal to the current analytical solvers.

### 4.3. Ablation Study

**Impact of Biased Weight Estimation.** To quantify the impact of the weight estimation mechanism discussed in Sec. 3.2, we conduct an ablation study focusing on its role during the denoising process. Specifically, we evaluate GOLDDIFF under two configurations: (1) employing the biased Weighted Streaming Softmax (WSS), and (2) utilizing the Unbiased Streaming Softmax (Dao et al., 2022) (SS). The results, reported in Tab. 6, demonstrate that biased weighting leads to performance degradation, whereas the unbiased formulation consistently yields superior denoising results. This is attributed to the efficacy of GOLDDIFF: by explicitly filtering a "Golden Subset" that is highly reliable and information-rich for the current noisy input, the unbiased estimator can effectively reconstruct the posterior.

**Hyperparameter Analysis.** We perform a sensitivity analysis on two critical hyperparameters: the maximum coarse subset size ($m_{max}$) and the minimum golden subset size ($k_{min}$). For *Coarse Subset Size ($m_{max}$)*: This parameter

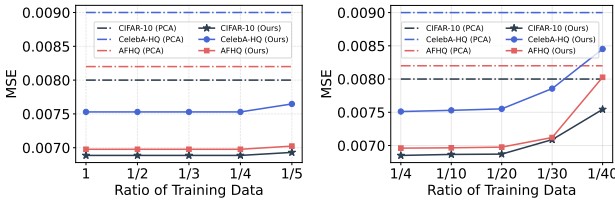

*(a)* Coarse Set $m_{max}$      *(b)* Golden Subset $k_{min}$

*Figure 6.* **Sensitivity Analysis** of the coarse candidate set size and golden subset size across multiple datasets. Dashed lines mean the baseline PCA (Lukoianov et al., 2025).

*Table 6.* Biased (WSS) vs. unbiased weight (SS) estimation. All metrics are averaged over 128 samples.

| | | Celaba-HQ | | AFHQ | |
|---|---|---|---|---|---|
| | | MSE ($\downarrow$) | $r^2$ ($\uparrow$) | MSE ($\downarrow$) | $r^2$ ($\uparrow$) |
| GOLDDIFF | + WSS | 0.009 | 0.818 | 0.008 | 0.715 |
| | +SS | **0.008** | **0.836** | **0.007** | **0.731** |

controls the late-stage denoising by determining the candidate pool size required to identify precise nearest neighbors. We evaluate $m_{max} \in \{1, 1/2, 1/3, 1/4, 1/5\}$ relative to the full training dataset size $N$. As illustrated in Fig. 6a, GOLDDIFF exhibits remarkable consistency across multiple datasets. To balance computational overhead and approximation accuracy, we empirically set $m_{max} = 4/N$ as the default. Notably, when $m_{max}$ is smaller, performance degrades significantly, as the subset may fail to encompass the optimal neighborhood for the subsequent golden subset selection. For *Golden Subset Size ($k_{min}$):* We further investigate the impact of the fine-subset selection bounds. We evaluate $k_{min}$ across the set $\{1/4, 1/10, 1/20, 1/30, 1/40\}$ relative to training data size $N$. We empirically set $k_{min} = 20/N$. The results in Fig. 6b demonstrate consistent performance across multiple datasets. Similar to the behavior observed with the coarse subset, a performance drop occurs at the extreme lower bound, where the subset becomes too sparse to provide sufficient local guidance for the diffusion process.

## 5. Conclusion

We resolved the scalability bottleneck of analytical diffusion by transforming a mathematically transparent yet prohibitive framework into a practical engine for large-scale generative modeling. Through the lens of *Posterior Progressive Concentration*, we proved that global full training set scanning is often redundant and even harmful. Our GOLDDIFF framework dynamically filters a *Golden Subset* via a theoretical-grounded dynamical selection, effectively decouples inference complexity from dataset scale. This approach achieves a $71\times$ speedup on AFHQ and represents the first successful scaling of analytical diffusion to ImageNet-1K. Furthermore, we establish a scalable training-free paradigm that bridges the gap between mathematical transparency and industrial-scale generation.

## Acknowledgments

This work is supported by the MBZUAI-WIS Joint Program for Artificial Intelligence Research.

## Impact Statement

This work may have positive societal impacts by improving the efficiency and transparency of diffusion models. The closed-form nature of analytical diffusion enhances interpretability by making explicit how individual training samples contribute to the generation process at different noise levels, which can support dataset attribution and a clearer scientific understanding of diffusion models. At the same time, the work may also have negative societal impacts. By making diffusion training cheaper and more efficient, it could lower the barrier to producing high-quality synthetic content, including deepfakes, misinformation, and non-consensual imagery. While analytical methods currently lag behind neural models in generation quality, which limits near-term misuse risk, this gap may narrow as the framework matures.

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

# Appendix

# A. Theoretical Analysis and Proofs

In this section, we provide the rigorous derivation of the approximation error bound presented in Thm. 1 and analyze the asymptotic behavior of our estimator in the high-noise and low-noise regimes.

### A.1. Proof of Theorem 1

**Problem Setup.** Recall that the exact analytical denoiser $\hat{\mathbf{f}}_{\mathcal{D}}(\mathbf{x}_t)$ is given by a softmax-weighted average over the entire dataset $\mathcal{D}$:

$$\hat{\mathbf{f}}_{\mathcal{D}}(\mathbf{x}_t) = \sum_{i=1}^{N} w_i \mathbf{x}_i, \quad w_i = \frac{\exp(\ell_i)}{Z}, \quad Z = \sum_{j=1}^{N} \exp(\ell_j), \tag{8}$$

where $\ell_i = -\|\mathbf{x}_t/\sqrt{\alpha_t} - \mathbf{x}_i\|^2/2\sigma_t^2$ are the logits, and $Z$ is the partition function.

Let $S_t$ be the subset of indices corresponding to the top-$k$ largest logits. Without loss of generality, assume the indices are sorted such that $\ell_{(1)} \geq \ell_{(2)} \geq \cdots \geq \ell_{(N)}$. Thus, $S_t = \{(1), \ldots, (k)\}$. The truncated estimator $\hat{\mathbf{f}}_{S_t}(\mathbf{x}_t)$ re-normalizes the weights over $S_t$:

$$\hat{\mathbf{f}}_{S_t}(\mathbf{x}_t) = \sum_{i \in S_t} \tilde{w}_i \mathbf{x}_i, \quad \tilde{w}_i = \frac{\exp(\ell_i)}{Z_S}, \quad Z_S = \sum_{j \in S_t} \exp(\ell_j), \tag{9}$$

where $Z_S$ is the partial partition function.

**Step 1: Decomposition of the Error.** The error vector is the difference between the exact and truncated estimates:

$$\mathbf{e} = \hat{\mathbf{f}}_{S_t}(\mathbf{x}_t) - \hat{\mathbf{f}}_{\mathcal{D}}(\mathbf{x}_t) \tag{10}$$

$$= \sum_{i \in S_t} \tilde{w}_i \mathbf{x}_i - \left( \sum_{i \in S_t} w_i \mathbf{x}_i + \sum_{i \notin S_t} w_i \mathbf{x}_i \right) \tag{11}$$

$$= \sum_{i \in S_t} (\tilde{w}_i - w_i) \mathbf{x}_i - \sum_{i \notin S_t} w_i \mathbf{x}_i. \tag{12}$$

**Step 2: Analyzing the Weight Difference.** Let $Z_{tail} = Z - Z_S = \sum_{j=k+1}^{N} \exp(\ell_{(j)})$ be the residual mass of the truncated samples. For any index $i \in S_t$, the relationship between the exact weight $w_i$ and truncated weight $\tilde{w}_i$ is:

$$w_i = \frac{e^{\ell_i}}{Z} = \frac{e^{\ell_i}}{Z_S + Z_{tail}} = \frac{e^{\ell_i}}{Z_S} \cdot \frac{Z_S}{Z_S + Z_{tail}} = \tilde{w}_i \left( 1 - \frac{Z_{tail}}{Z} \right). \tag{13}$$

Thus, the difference is:

$$\tilde{w}_i - w_i = \tilde{w}_i - \tilde{w}_i \left( 1 - \frac{Z_{tail}}{Z} \right) = \tilde{w}_i \frac{Z_{tail}}{Z}. \tag{14}$$

Note that this difference is always positive, as renormalization increases the weights of retained samples.

**Step 3: Bounding the Norm.** Applying the triangle inequality to the error vector $\mathbf{e}$:

$$\|\mathbf{e}\|_2 \leq \sum_{i \in S_t} |\tilde{w}_i - w_i| \|\mathbf{x}_i\|_2 + \sum_{i \notin S_t} |w_i| \|\mathbf{x}_i\|_2 \tag{15}$$

$$\leq R \left( \sum_{i \in S_t} (\tilde{w}_i - w_i) + \sum_{i \notin S_t} w_i \right), \tag{16}$$

where $R = \max_i \|\mathbf{x}_i\|_2$ is the bound on the data norm. Substituting the expressions from Step 2:

- First term sum: $\sum_{i \in S_t} (\tilde{w}_i - w_i) = \sum_{i \in S_t} \tilde{w}_i \frac{Z_{tail}}{Z} = \frac{Z_{tail}}{Z} \underbrace{\sum_{i \in S_t} \tilde{w}_i}_{1} = \frac{Z_{tail}}{Z}.$

- Second term sum: $\sum_{i \notin S_t} w_i = \frac{Z_{tail}}{Z}$ (by definition of $Z_{tail}$).

Thus, the total error is bounded by:

$$\|\mathbf{e}\|_2 \le R \left( \frac{Z_{tail}}{Z} + \frac{Z_{tail}}{Z} \right) = 2R \frac{Z_{tail}}{Z}. \tag{17}$$

**Step 4: Bounding the Ratio with Logit Gap.** We now bound the ratio $Z_{tail}/Z$ using the sorted logits.

- Lower bound for $Z$: Since $\ell_{(1)}$ is the maximum logit, $Z > \exp(\ell_{(1)})$.

- Upper bound for $Z_{tail}$: The tail consists of $N - k$ samples, each with logit $\ell_{(j)} \le \ell_{(k+1)}$ for $j > k$. Thus, $Z_{tail} = \sum_{j=k+1}^{N} e^{\ell_{(j)}} \le (N - k)e^{\ell_{(k+1)}}$.

Substituting these into Eq. (17):

$$\frac{Z_{tail}}{Z} \le \frac{(N - k)e^{\ell_{(k+1)}}}{e^{\ell_{(1)}}} = (N - k) \exp\left( -(\ell_{(1)} - \ell_{(k+1)}) \right). \tag{18}$$

Defining the Logit Gap as $\Delta_k \triangleq \ell_{(1)} - \ell_{(k+1)}$, we obtain the final bound:

$$\|\hat{\mathbf{f}}_{\mathcal{D}}(\mathbf{x}_t) - \hat{\mathbf{f}}_{S_t}(\mathbf{x}_t)\|_2 \le 2R(N - k) \exp(-\Delta_k). \tag{19}$$

$\square$

### A.2. Asymptotic Analysis of Regime Dynamics

Here, we analyze the behavior of the Logit Gap $\Delta_k$ as a function of the noise level $\sigma_t$, providing the mathematical justification for the "Integration-to-Selection" phase transition discussed in Sec. 3.

Let the distance between the noisy query $\mathbf{x}_t$ and a training sample $\mathbf{x}_i$ be $d_i = \|\mathbf{x}_t/\sqrt{\alpha_t} - \mathbf{x}_i\|^2$. The logit is given by $\ell_i = -d_i/2\sigma_t^2$. The Logit Gap becomes:

$$\Delta_k(\sigma_t) = \frac{d_{(k+1)} - d_{(1)}}{2\sigma_t^2}. \tag{20}$$

Let $\delta_k = d_{(k+1)} - d_{(1)} > 0$ be the raw distance gap between the top-1 and the $(k + 1)$-th neighbor.

**Case 1: High-Noise Regime ($\sigma_t^2 \to \infty$).** In the early stages of reverse diffusion, $\sigma_t^2$ is very large.

$$\lim_{\sigma_t^2 \to \infty} \Delta_k(\sigma_t) = \lim_{\sigma_t^2 \to \infty} \frac{\delta_k}{2\sigma_t^2} = 0. \tag{21}$$

Consequently, the exponential decay term $\exp(-\Delta_k) \to e^0 = 1$. The error bound simplifies to:

$$\text{Error} \lesssim 2R(N - k). \tag{22}$$

**Implication:** The error is linearly proportional to the number of discarded samples $(N - k)$. To minimize error, we must minimize $(N - k)$, which implies maximizing $k$ (i.e., $k \to N$). This proves that *dense aggregation* is mathematically necessary in the high-noise regime.

**Case 2: Low-Noise Regime ($\sigma_t^2 \to 0$).** As the diffusion process nears the data manifold, $\sigma_t^2 \to 0$. Assuming the query $\mathbf{x}_t$ is not equidistant to $\mathbf{x}_{(1)}$ and $\mathbf{x}_{(k+1)}$ (which holds almost surely for continuous data), we have $\delta_k > 0$.

$$\lim_{\sigma_t^2 \to 0} \Delta_k(\sigma_t) = \lim_{\sigma_t^2 \to 0} \frac{\delta_k}{2\sigma_t^2} = +\infty. \tag{23}$$

Consequently, the exponential term vanishes rapidly:

$$\lim_{\sigma_t^2 \to 0} \exp(-\Delta_k) = 0. \tag{24}$$

Crucially, this decay is exponential in $1/\sigma_t^2$, which dominates the linear term $(N - k)$. Even if we discard the vast majority of the dataset (i.e., $N - k \approx N$, $k$ is small), the total error remains negligible. This proves that *sparse selection* is theoretically sufficient in the low-noise regime.

*Table 7.* **Quantitative Comparison of Analytical Denoisers on MNIST and Fashion-MNIST.** *Time* (s) is averaged over 128 samples. Best and second-best results are **bolded** and underlined, respectively.

| Dataset | MNIST | | | Fashion-MNIST | | |
|---|---|---|---|---|---|---|
| Method | MSE ($\downarrow$) | $r^2$ ($\uparrow$) | Time | MSE ($\downarrow$) | $r^2$ ($\uparrow$) | Time |
| Optimal (De Bortoli, 2022) | 0.047 | 0.435 | 2.806 | 0.033 | 0.514 | 2.608 |
| Wiener (Wiener, 1949) | 0.034 | 0.594 | 0.001 | 0.031 | 0.589 | 0.001 |
| Kamb (Kamb & Ganguli, 2024) | 0.043 | 0.489 | 4.115 | 0.051 | 0.294 | 3.903 |
| PCA (Lukoianov et al., 2025) | 0.027 | 0.684 | 2.798 | 0.015 | 0.795 | 2.791 |
| GOLDDIFF (Ours) | **0.020** | **0.713** | **0.019** | **0.011** | **0.813** | **0.088** |

## A.3. Extension to Localized Estimators (PCA)

**Corollary 1 (Sample-wise Error Bound for Local Estimators) .** *Consider a localized estimator (e.g., PCA Denoiser (Lukoianov et al., 2025)) that computes the denoising score independently for each spatial location (or patch) $p$:*

$$\hat{\mathbf{f}}^{(p)}(\mathbf{x}_t) = \sum_{i=1}^{N} w_i^{(p)} \mathbf{x}_i^{(p)}, \quad w_i^{(p)} \propto \exp\left(-\|\mathbf{x}_t^{(p)} - \mathbf{x}_i^{(p)}\|^2 / 2\sigma_t^2\right). \tag{25}$$

*Let $S_t$ be a globally selected subset. The approximation error at location $p$ is bounded by:*

$$\|\hat{\mathbf{f}}_{\mathcal{D}}^{(p)} - \hat{\mathbf{f}}_{S_t}^{(p)}\|_2 \le 2R^{(p)}(N - |S_t|)\exp\left(-\Delta_k^{(p)}\right) + \epsilon_{mismatch}, \tag{26}$$

*where $\Delta_k^{(p)}$ is the local Logit Gap at position $p$.*

**Remark.** Thm. 1 applies *sample-wise* to each spatial location. The error consists of two parts:

- Truncation Error (The Exponential Term): This follows directly from Thm. 1. In the low-noise regime, the local Logit Gap $\Delta_k^{(p)}$ explodes, driving this error to zero. This confirms that *if* the true local neighbors are present in $S_t$, the PCA estimator converges.

- Selection Mismatch Error ($\epsilon_{\text{mismatch}}$): This term arises if the globally selected subset $S_t$ fails to include the true local top-$k$ neighbors for position $p$. However, as justified in Sec. 3.4, natural images exhibit strong hierarchical consistency. A sample that is a "true neighbor" at a local patch level typically exhibits high similarity in the global low-frequency proxy used for selection. Furthermore, by maintaining a sufficiently large candidate pool $m_t$, we minimize the probability of exclusion ($\epsilon_{\text{mismatch}} \approx 0$).

# B. Additional Experiments

**Results of MNIST and Fashion-MNIST.** We evaluate the performance of GOLDDIFF against established analytical baselines across MNIST and Fashion-MNIST. The results are reported in Tab. 7. Obviously, our GOLDDIFF consistently surpasses the state-of-the-art PCA method (Lukoianov et al., 2025) across all MSE and $r^2$ metrics with significant acceleration. Qualitatively, Fig. 4 shows that GOLDDIFF exhibits superior similarity to the ground-truth neural diffusion outputs (see last row) on these datasets.

**Computation of Proxy Space $\mathbb{R}^d$ And Choice of $d$.** At each step, both $\mathbf{x}_t$ and all training samples $\{\mathbf{x}_i\}$ are downsampled ($D \to d = D/16$) via average pooling. $\ell_2$ distances are then computed in this space to form $\mathcal{C}_t$, from which the final golden subset $S_t$ of size $k_t$ is selected using full-resolution distances (Eq. 5).

Specifically, we conduct experiments on ImageNet-1K to verify the validity of our default choice $d = D/16$, where $D$ is the full original dimension. We vary $d$ from $D$ (no downsampling) to $D/256$ and report MSE and $r^2$. As shown in Tab. 8, performance remains nearly unchanged from $d=D$ down to $d=D/16$, and only degrades slightly beyond that. This confirms that $d=D/16$ achieves an effective balance between efficiency and quality.

Then, we verify the reliability of the low-dimensional proxy, which is designed to reduce cost by first providing a coarse

candidate set $\mathcal{C}_t$, from which the final top-$k_t$ subset is selected (Eq. 5). We measure the proportion of true top-$k_t$ neighbours (identified in the full dimension $D$) that are contained in $\mathcal{C}_t$, focusing on the low-noise regime ($t \leq 200$) where retrieval precision is critical. As shown in Tab. 8, with $d = D/16$, all true top-$k_t$ neighbours are fully contained, confirming that the proxy introduces no retrieval loss.

*Table 8.* Performance and retrieval reliability under varying proxy dimension $d$ on ImageNet-1K.

| $d$ | MSE ($\downarrow$) | $r^2$ ($\uparrow$) | top-$k_t \subseteq \mathcal{C}_t$ |
|---|---|---|---|
| $D$ (none) | 0.031 | 0.494 | 100% |
| $D/4$ | 0.031 | 0.493 | 100% |
| $D/16$ (default) | 0.031 | 0.490 | 100% |
| $D/64$ | 0.032 | 0.485 | 99.3% |
| $D/256$ | 0.033 | 0.480 | 98.3% |

**Practical Applications Beyond Explainability.** One promising use is employing the analytical denoiser as a supervision signal for neural denoiser training. The analytical posterior mean $\mathbb{E}[\mathbf{x}_0|\mathbf{x}_t]$ is a weighted average over the training set, which provides a low-variance training target. As a proof-of-concept, we conduct an experiment on MNIST using CNN, varying only the training target: 'S' uses clean sample $\mathbf{x}_0$, while 'A' uses the closed-form posterior mean. Tab. 9 verifies that analytical supervision yields *faster convergence* (lower training loss) and *better generation quality* (lower FID). Scaling to larger datasets and architectures is a promising future direction.

*Table 9.* Standard ('S') vs. analytical ('A') supervision.

| Epoch | FID (S) | FID (A) | Loss (S) | Loss (A) |
|---|---|---|---|---|
| 1 | 263.7 | **252.5** | 0.037 | **0.013** |
| 10 | 43.4 | **39.1** | 0.030 | **0.008** |
| 30 | 20.2 | **13.4** | 0.021 | **0.005** |

