# OpenReview forum: "Fast and Scalable Analytical Diffusion"
_ICML.cc/2026/Conference — ICML 2026 regular_

### Official Review · Reviewer_poUR · 2026-02-18

**Soundness:** 2
**Presentation:** 3
**Significance:** 2
**Originality:** 3
**Overall Recommendation:** 3
**Confidence:** 3

**Summary:**

This paper addresses the scalability problem in analytical diffusion, where empirical-bayes denoisers must scan the whole dataset at each timesteps. The key idea is Posterior Progressive Concentration, meaning the posterior support shrink as noise decreases. Based on this, the authors propose a training-free method called Dynamic Time-Aware Golden Subset Diffusion (GOLDIFF). It uses a time-varying retrieval schedule to approximate the empirical-bayes score and provides truncation error bounds. The method achieves notable speedups, like being 71 times faster on AFHQ, and scales to ImageNet-1K.

**Compliance With Llm Reviewing Policy:**

Affirmed.

**Final Justification:**

They partially addressed my concerns.

**Key Questions For Authors:**

SWee the weaknesses above

**Limitations:**

Yes

**Strengths And Weaknesses:**

Strengths

1. The paper points out a useful phenomenon: posterior support shrinks as the signal-to-noise ratio increases. It turns this into a practical inference-time algorithm with clear schedule for candidate set size and aggregation size. Empirically, the method shows significant improvements while maintaining or even improving alignment with a neural “oracle” using metrics like $\mathrm{MSE}$ and $r^2$. The scaling result on Imagenet-1K is also encouraging for analytical diffusion.


Weaknesses

1. The claim that the method is “training-free” and independent of dataset size seems overstated because there is still some dependence on the dataset. Even with GOLDIFF, the stated complexity still includes an (O(Nd)) term (see Table 1). The method filters the dataset into a candidate set $C_t$ using a proxy distance at each timestep $t$, which implies scaning all $N$ samples unless an indexing method is applied. If the method depends on approxamate nearest neighbor techniques like FAISS or precomputed featurs, this should be clearly mentioned since it affects memory use, preprocessing time, and reproducibility.

2. The proxy retrieval approach is heuristic, and its potential failure modes are not well explained.
The coarse screening uses downsampled ($\ell_2$ distance at a scale of $s=\tfrac{1}{4}$ to build $C_t$, based on the idea of “hierarchical consistency.” However, it’s unclear how reliable this is when low-fq similarity can be misleading, such as with textures, fine-grained categories, or some medical images. The schedule increases $m_t$ as noise decreases to provide a “safety margin,” but there’s no strong analysis on how likely it is that the true nearest neighbours (according to the real score weighting) stay within $C_t$, especially in the low-noise range where selection matters most.

3. The subset sizes chosen seem quit large at scale, which might limit the actual speed improvments.
The implementation sets

$m_{\min} = k_{\max} = \frac{N}{10}, \qquad m_{\max}=\frac{N}{4}, \qquad k_{\min}=\frac{N}{20}$.

For large $N$, values like $\tfrac{N}{10}$ or $\tfrac{N}{4}$ are still very large, so the “subset” isn’t very sparse. This raises the question of whether the speedups come partly from inefficiencies in the baseline methods (like PCA weight estimation) rather than the subset approach alone. It also makes me wonder if similar speedups would appear with optimized baselines or ANN-based retrieval methods.

4. A mathematical concern is that the truncation error bound might be loose and somewhat circular when used to guide the design.
Theorem 1 bounds the truncation error in score estimation, written as something like
$
\bigl\lVert \hat f_D(x_t) - \hat f_{S_t}(x_t) \bigr\rVert_2,
$
Here, $\hat f_D(x_t)$ is the full-dataset estimator and $\hat f_{S_t}(x_t)$ is the subset estimator. The bound depends on factors like a radius term, the tail size $N-k$, and a gap term involving $\exp(-\Delta_k)$, where $\Delta_k$ is the “logit gap.” In practice, the bound can become meaningless if R is large or $\Delta_k$ isn’t well controlled. Also, accurately evaluating $\Delta_k$ requires knowing the top $k+1$ logits, which depends on a reliable retrieval step. I would like to see more evidence that this bound is tight enough to predict actual score error, such as correlation plots between $\Delta_k$ and $\lVert \hat f_D(x_t)-\hat f_{S_t}(x_t)\rVert_2$.

5. The evaluation rely on an oracle-alignment proxy that might not accurately reflect the true quality of the samples.
Much of the quantitative evaluation compares an analytical estimator to a neural denoiser “oracle” using
$
\mathrm{MSE} \quad \text{and} \quad r^2.
$
This measures agreement with a specific learned model, not necesarily sample fidelity under standard generative metrics (FID, Inception Score, precision/recall). The paper shows qualitative grids, but given the claim of improved generation and scalability, I expected more standard quality metrics and clear sampling setup (conditioning, guidance, class-conditional protocol).

6. It’s difficult to judge baseline fairness because multiple components are intertwined. The comparison with PCA variants involves several factors: (a) subset selection, (b) weight estimation using streaming softmax vs. other methods, and (c) the projection operator. The ablation study separating WSS from SS is useful, but it remains hard to tell how much of the improvement comes from the time-varying schedules ${m_t}), ({k_t}$ versus using a different estimator on a better support set.

7. If GOLDIFF needs storing per-sample downsampled representations, patch features, or an index over all $N$ training points, memory and preprocesing can dominate at Imagenet scale. The paper reports peak memory per step, but not the end-to-end footprint including cached features or indexing (if any). For a claim of scaling to Imagenet-1K, this engineering detail matters for reproducibility.

---

> ### Author Rebuttal · Authors · 2026-03-31
>
> We appreciate your constructive comments, which are definitely helpful to improve the quality of our paper. All suggestions will be accommodated. Below, we provide detailed clarifications to each of your questions.
> >**W1: Clarification of training-free, complexity, and implementation details.**
> - **Training-free** means no neural network training, learned parameters, or backpropagation.
> - **Complexity.** We clarify that our claim refers to the score computation stage: prior methods compute the posterior over all training set, whereas we use only a small golden subset. We will clarify this in the revision.
> - **Implementation.** GoldDiff uses **no FAISS** or indexing structure. We will add details and release the codebase.
>
> >**W2: Proxy retrieval reliability.**
>
> Thanks for the suggestion. We clarify that **(i) Design rationale.** The proxy only needs to *not miss* true neighbours; final top-$k_t$ selection uses full-resolution distances, and the safety margin ensures high recall. **(ii) Empirical validation.** Containment of true top-$k_t$ neighbours in $\mathcal{C}_t$ reaches 100% (see Table R3 to Reviewer bbPD [Q1] for full results). Additionally, Spearman $\rho$ between proxy and full-resolution distances reaches 0.99 at low noise on fine-grained ImageNet-1K (Table R7). CelebA-HQ shows the same trend ($\rho{:}\ 0.935 \to 0.996$), confirming that the proxy is reliable.
>
> Table R7: Spearman $\rho$.
> |$t$|0.9|0.5|0.2|0.1|
> |---|:---:|:---:|:---:|:---:|
> |$\rho$|0.910|0.985|0.990|0.991|
> >**W3: Subset sizes and source of improvement.**
>
> Thanks for the comment. We clarify that **(i) Subset sizes.**  $K_{\max}$ is not large in practice, especially since competing methods use the full set, while our method still achieves a significant speedup. Furthermore, at low noise, $k_t$ shrinks to $N/20$ and coarse screening operates in $D/16$ proxy space, yielding **71$\times$** end-to-end speedup on AFHQ (Table 2). **(ii) Source of improvement.** Ablation (Table R9 in [W6]) fixes SS throughout: a static subset \(C\) underperforms full-scan (B) ($r^2$: 0.461 vs. 0.473), while the dynamic schedule \(G\) achieves the best result (0.490). This confirms that the gains mainly originate from time-varying schedule, which ANN-based retrieval cannot provide.
>
> >**W4: Tightness and circularity of the truncation error bound.**
>
> Thanks for the question. We clarify: **(i) No circularity.** Theorem 1 is not used at runtime and only provides post-hoc justification. Schedule (Eq. 6) is a closed-form function of $t$, fully determined before inference. **(ii) $R$ and $\Delta_k$ do not make the bound meaningless.** With images normalized to $[-1,1]^D$, $R \leq \sqrt{D}$; on ImageNet-1K, the measured $R{=}102.7$. As $\Delta_k$ grows rapidly (Table R8), $\exp(-\Delta_k) \approx 0$ and the bound vanishes. **(iii) Empirical tightness.** At $t{=}0.6$ the bound (7.8e-5) is within 6× of actual error (1.3e-5); by $t{=}0.4$ both vanish (Table R8).
>
> Table R8. Bound tightness on ImageNet-1K.
> |$t$|$\Delta_k$|Bound|Actual Error|
> |---|:---:|:---:|:---:|
> |0.6|21.9|7.8e-5|1.3e-5|
> |0.4|145.6|≈0|≈0|
> |0.2|1027|≈0|≈0|
> |0.1|4268|≈0|≈0|
>
> >**W5: Rationale for MSE & $r^2$ and FID.**
>
> Thanks for the suggestion. We clarify that the primary goal of analytical diffusion is to interpret neural diffusion models, thus all prior works compare analytical outputs against neural ones via MSE and $r^2$. Following your suggestion, we additionally provide FID evaluation: ***GoldDiff achieves consistently lower FID than PCA across datasets*** (see Table R2 in response to Reviewer bbPD [W1] for full results). As for the sampling setup, all methods use the same initial noise and 10 DDIM steps; class-conditional results (ImageNet-1K) are generated per-class. We will add details in the revision.
>
> >**W6: Ablation.**
>
> Following your suggestion, we conduct a full ablation on ImageNet-1K. In Table R9, static ('S') uses the midpoint
> of the dynamic ('D') range for fair comparison. The results show that:
> - (a) Subset selection ($m_t$, $k_t$): Dynamic $m_t$ (C→D) and dynamic $k_t$ (C→E) each improve over the static baseline; combining them (G) achieves the best result.
> - (b) Weight estimation: Replacing WSS with SS yields consistent gains on both the full dataset (A→B) and the golden subset (F→G).
> - \(c\) Projection operator: GoldDiff is plug-and-play and independent of the projection. Table 5 confirms this.
>
> Table R9. Ablation. '—' means Full data.
> ||$m_t$|$k_t$|Estimator|MSE|$r^2$|
> |---|:---:|:---:|:---:|:---:|:---:|
> |A|—|—|WSS|0.033|0.467|
> |B|—|—|SS|0.032|0.473|
> |C|S|S|SS|0.033|0.461|
> |D|D|S|SS|0.032|0.484|
> |E|S|D|SS|0.032|0.480|
> |F|D|D|WSS|0.032|0.486|
> |**G**|**D**|**D**|**SS**|**0.031**|**0.490**|
>
> >**W7: End-to-end footprint.**
>
> Thanks for the question. We clarify that the reported peak memory and inference time in Table 2 are end-to-end: GPU memory is measured via `max_memory_allocated` after the full database (including downsampled copy) is loaded, and time covers all data processing.

---

> > ### Author Rebuttal · Reviewer_poUR · 2026-04-02
> >
> > Thank you to the authors for the detailed and data-driven rebuttal. You have addressed several of my specific technical concerns, particularly by providing the Spearman correlations for proxy reliability, the empirical tightness of the truncation bound, and the requested FID scores. The ablation in Table R9 also effectively isolates the gains of the time-varying schedule.
> >
> > However, while the rebuttal strengthen the empyrical standing of the paper, I still have lingering reservations regarding the core claims of true "scalability" and being "independent of dataset size":
> >
> > * **Scalability and $\mathcal{O}(Nd)$ Overhead:** As clarified, the method still requires computing distances against a downsampled proxy of the *entire* dataset at every single step. For ImageNet-1K, your default $m_{max} = N/4$ means retaining and processing 250,000 samples for the coarse set. Because you are not utilizing indexing structurs like FAISS, this $\mathcal{O}(Nd)$ overhead is still fundamentally bottlenecked by $N$. Therefore, the claim that GOLDDIFF "decoupeles inference complexity from dataset size" feels overstated.
> > * **Theoretical Bound:** While Table R8 shows the bound is non-vacuous at lower noise levels, the bound remains a post-hoc justification rather than the rigorous driver of the heuristic scheduling.
> >
> > Overall, the empirical results are interesting, and the dynamic schedule is a clever improvement over standard PCA weight estimation. However, the fundamental scalability limitations remain heavily tied to $N$. Because the merits are clearer but the broader scalability claims remain partially unconvincing, I am raising my score from a 2 to a 3.
> >
> > I will raise my mark to 3.

---

> > > ### Author Response · Authors · 2026-04-07
> > >
> > > We sincerely thank the reviewer for the constructive engagement throughout this discussion. Some of the further comments appear to be based on minor biases and can be easily clarified. Nevertheless, your suggestions have been invaluable in helping us strengthen both the paper and the theoretical/experimental solidness. We further clarify the two follow-up questions below.
> > >
> > > >**F1: Scalability and $\mathcal{O}(Nd)$ overhead.** The method still requires $\mathcal{O}(Nd)$ coarse screening against the entire dataset at every step. FAISS is not used, and the claim of decoupling from $N$ feels overstated.
> > >
> > > Thanks for raising this. We clarify that while coarse screening involves $N$, profiling reveals that **it accounts for less than 2% of total inference time on large-scale datasets**. Specifically, we decompose GoldDiff into three per-step stages: (1) coarse screening in $\mathbb{R}^d$ ($d{=}D/16$), (2) fine selection within the coarse set, and (3) score computation on the golden subset. Profiling reveals that score computation dominates at 90–95% of total time on high-resolution data, while coarse screening accounts for < 2%. Prior methods spend 100% on the expensive score estimation over all $N$ samples ($\mathcal{O}(Np_tD)$ per step). GoldDiff reduces this dominant cost to $\mathcal{O}(k_tp_tD)$ with $k_t \ll N$, achieving 17 ×-71 × speedups. Moreover, the coarse overhead decreases with resolution, dropping from 10.2% (CIFAR-10) to < 2% (high-resolution datasets), verifying that **scalability is not a practical limitation**.
> > >
> > > |Dataset|PCA(s)|GoldDiff(s)|Speedup|Coarse %|Fine %|Score %|
> > > |---|:---:|:---:|:---:|:---:|:---:|:---:|
> > > |CIFAR-10|2.802|0.087|32 ×|10.2|26.2|63.6|
> > > |CelebA-HQ|6.040|0.349|17 ×|1.1|4.2|94.7|
> > > |AFHQ|24.896|0.351|71 ×|0.7|4.3|95.0|
> > > |ImageNet-1K|110.798|2.607|42 ×|1.7|8.1|90.3|
> > >
> > > We further clarify two points:
> > > - **On FAISS.** We clarify that standard ANN indices are ***not directly applicable*** here because coarse screening distance involves a time-varying coefficient $\alpha_t$ that changes at every denoising step, invalidating any pre-built index.
> > > - **On the scalability claim.** The original statement aims to emphasize that GoldDiff decouples the score computation stage from the dataset size. We will make it more precise and revise it to: "GoldDiff significantly reduces the dominant $\mathcal{O}(Np_tD)$ score computation to $\mathcal{O}(k_tp_tD)$ with $k_t \ll N$."
> > >
> > > > **F2: Theoretical Bound:** The bound remains a post-hoc justification rather than the rigorous driver of the heuristic scheduling.
> > >
> > > Thanks for this comment. We first clarify that Theorem 1 reveals the structural mechanism underlying the schedule: the truncation error is jointly controlled by $(N{-}k)$ and $\exp(-\Delta_k)$, which exhibit opposing behaviors across noise regimes. This explains *why* the counter-monotonic schedule is the right design choice. The bound is also non-vacuous at lower noise levels (Table R8), confirming that it provides meaningful quantitative guarantees.
> > >
> > > Furthermore, the pipeline of our work follows a prevalent paradigm in the diffusion model literature: empirical insight (Posterior Progressive Concentration, Sec. 3.1) $\to$ algorithm design (counter-monotonic schedule, Eq. 4 & Eq. 6) $\to$ theoretical guarantee (Theorem 1 bounds the truncation error). For example, the cosine noise schedule [1] was designed as a heuristic and only later shown to be Fisher-Information-motivated [2]; Classifier-free guidance [3] was proposed without theoretical guarantees, with foundations emerging only recently [4, 5]. In these cases, the theory is widely regarded as a meaningful contribution because it explains why the design works, which is the same role Theorem 1 serves in our work.
> > >
> > > ---
> > > **References:**
> > >
> > > [1] Improved Denoising Diffusion Probabilistic Models, ICML 2021.
> > >
> > > [2] Using Ornstein-Uhlenbeck Process to understand Denoising Diffusion Probabilistic Model and its Noise Schedules, 2023.
> > >
> > > [3] Classifier-Free Diffusion Guidance, NeurIPS 2021 Workshop.
> > >
> > > [4] Classifier-Free Guidance is a Predictor-Corrector, TMLR 2025.
> > >
> > > [5] Towards Understanding the Mechanisms of Classifier-Free Guidance, NeurIPS 2025.
> > >
> > > >**Overall:** The empirical results are interesting, and the dynamic schedule is a clever improvement. Broader scalability claims remain partially unconvincing.
> > >
> > > Thanks for your recognition and encouraging comment. Fundamental scalability is in fact a key strength of our work rather than a limitation. As we have explained above, our method only requires about **2% computation** to traverse $N$ on large-scale datasets, which is essentially negligible and much more efficient than any other counterparts. Therefore, this concern does not really arise in practice. Nevertheless, thank you again for your constructive comments and suggestions in the review. We will carefully revise our paper accordingly.

---

### Official Review · Reviewer_kdh2 · 2026-02-28

**Soundness:** 3
**Presentation:** 3
**Significance:** 2
**Originality:** 2
**Overall Recommendation:** 3
**Confidence:** 4

**Summary:**

The paper proposes an optimized analytical diffusion method. Instead of requiring the full dataset during inference, the authors exploit the posterior progressive concentration property inherent in diffusion models that allows for focusing on small subsets. The proposed method, dubbed GoldDiff, dynamically selects such subsets via a coarse-to-fine mechanism. Empirical results are given for several benchmarks: (F-)MNIST, CIFAR-10, CelebA-HQ, AFHQ and, most notably, ImageNet-1k, for which full-data analytical methods are infeasible.

**Compliance With Llm Reviewing Policy:**

Affirmed.

**Final Justification:**

The rebuttal reinforced my prior assessment, I have therefore maintained my original score and I am increasing my confidence.

The paper is on analytical methods for diffusion models, a topic which is of interest in the context of interpretability and explainability.

Fundamental concerns that remain unaddressed are as follows:

I originally mentioned that the paper heavily focuses on benchmarking and offers little insight into the inner workings of diffusion models. The main point of this work is to accelerate, scale analytical methods to larger setups and the writing explicitly highlights this as the central contribution. However, it is unclear why one would be interested in large scale analytical methods and in my opinion this distracts from their main purpose.

During discussion, the authors have tried to directly address the above by adding some interpretability context. While this could be promising, I did not find it entirely convincing: given the paper's framing it felt rushed and more like an afterthought.

The authors have also tried another angle to justify their approach: a way to accelerate diffusion model training. They claim 25% improvement in FID on MNIST. However, replacing denoising score matching with analytical scores reduces optimization noise so the results are not surprising. Moreover, the benefits of GoldDiff analytical scores versus the theoretical scores also remain unexplored here. In any case, deciding the merit of the submission based on new angles of interpretation, not present in the paper, is not appropriate.

Lastly, focusing on the core of the paper, I find that some claims are not well-supported and evidence would need to be strengthened. For example, all experiments use 128-512 random samples to assess efficacy and conclude the superiority of GoldDiff. However, especially on the scale of ImageNet, which has over a million samples, this is not enough as it does not even cover the number of distinct classes. For a paper that advertises new scalability results this is important.

**Key Questions For Authors:**

1. The writing starting line 200, column 2 is unclear to me. How do you select $\mathcal{C}_t$? What do $\boldsymbol{x}_i$ and $\boldsymbol{x}_t$ stand for? I assume the former is a sample from $\mathcal{D}$ while the latter is obtained during inference? Do you threshold $d^{\text{proxy}}$? Could the authors include an algorithm snippet of the full method to clarify?

2. As I understand, the comparisons are with respect to a convolutional neural denoiser, to see whether analytical methods can approximate what is being learned. Moreover, it seems that the proposal is to reweight (sparsely) the optimal denoiser. Given that the progressive concentration property is architecture agnostic (it follows from the theoretical solution in Equation 2), why should we expect GoldDiff to track with learning biases of convolutional models or natural images?

[1] Kamb et al., *An analytic theory of creativity in convolutional diffusion models*, ICML 2025

**Limitations:**

The paper does not explicitly discuss limitations. I encourage the authors to include a Limitations section in the final version of the paper. At present, I believe that a major limitation, also mentioned in W2 above, is that GoldDiff does not yield any new insights into neural diffusion models.

**Strengths And Weaknesses:**

Strengths
-

1. Overall, the paper is well-written and easy to follow.
2. The proposed method is extensively validated.
3. GoldDiff is shown to lead to faster inference and comparable performance to previous analytical methods.


Weaknesses
-

1. While I acknowledge the benefits of the proposed method, especially in inference times, in my opinion, the novelty is somewhat incremental. The progressive concentration property is a well-known consequence of the theory (Equation 2) and the reported benefits in efficacy are marginal, e.g., see Table 2 MSE especially.

2. Most importantly, the paper's focus on benchmarks distracts from the main purpose of analytical methods, which is interpretability. For example, [1], which is well-cited in the paper, gives insight into specific inductive biases relating to locality. It remains unclear to me how progressive concentration relates to neural biases here.

---

> ### Author Rebuttal · Authors · 2026-03-31
>
> Thanks for your thorough and constructive comments. We address your concerns regarding novelty, efficacy, and interpretability, and highlight new insights for neural denoisers. All the suggestions will be included in our revised paper. Below, we provide detailed responses for each of your questions:
> >**W1-1: The progressive concentration property.**
>
> Thanks for your kind comment. We would like to clarify that our contribution goes beyond the observation that softmax concentrates at low noise, and we are confident in the originality and significance of our work (other reviewers acknowledged our approach is "original and pretty smart"). Specifically:
> - (i) We provide the ***first*** systematic characterization of the two-regime dynamics across the diffusion trajectory (Sec. 3.3, Fig. 3).
> - (ii) We reveal a ***counter-intuitive finding*** that full-dataset scanning is not merely wasteful but actively harmful, in contrast, a subset-based unbiased estimator surpasses the full-scan baseline (Table 2). Moreover, we identify that one bottleneck of prior SOTA lies in the estimation bias (Sec. 3.2).
> - (iii) We translate these insights into a ***complete pipeline***: counter-monotonic schedule (Sec. 3.4) + truncation error bound (Theorem 1), achieving 71$\times$ speedup, the best performance so far, and the first scaling to ImageNet-1K.
>
> >**W1-2: Efficacy.**
>
> We clarify that MSE alone does not capture the full picture:
> - (i) **Simultaneous speedup and quality improvement.** GoldDiff consistently outperforms the full-scan PCA baseline in both MSE and $r^2$ (Table 2), while being 71$\times$ faster on AFHQ.
> - (ii) **Generality.** GoldDiff is a plug-and-play module that improves different methods (Table 5).
> - (iii) **Scalability.** GoldDiff enables the first successful scaling to ImageNet-1K (Table 3).
> - (iv) **Practicality.** We demonstrate that the analytical posterior mean can serve as a low-variance supervision signal for neural diffusion training, yielding 25% lower FID (13.4 vs. 20.2 on MNIST; see Table R1 in response to Reviewer ducc [Q6] for full results).
>
> >**W2 and Q2: Interpretability of GoldDiff and Architecture-agnosticism.**
>
> Thanks for the question. We clarify that the generalization ability of diffusion models is shaped by two factors: ***architectural bias*** (e.g., CNN locality) and ***data bias*** (intrinsic statistical properties). [1] analyzes the former; however, PCA (Lukoianov et al., 2025) shows that the locality phenomenon arises from the data's intrinsic structure,
> positioning data bias as the more fundamental perspective. ***GoldDiff provides a new interpretability from data bias***: while PCA reveals a static bias, i.e., which components matter for denoising, GoldDiff reveals a dynamic dataset-dimension bias, i.e., which training samples contribute to the score and how this support set evolves with the noise level.
>
> Architecture-agnosticism is *precisely a strength*: since GoldDiff captures a data-driven property, it naturally holds across architectures, as verified by Table 4.
>
> >**Q1: Algorithm clarity.**
>
> As suggested, we provide the algorithm snippet of our GoldDiff below and will include it in the revision:
>
> **Input:** denoising step $t$, noisy sample $\mathbf{x}\_t$, training set $\mathcal{D} = \\{\mathbf{x}\_i\\}\_{i=1}^N$, schedules $m_t$ (Eq. 4), $k_t$ (Eq. 6)
> 1. **Coarse screening:** Downsample all samples ($D \to d{=}D/16$) via average pooling. Compute proxy $\ell_2$ distances. Select top-$m_t$ nearest → $\mathcal{C}_t$.
> 2. **Precision selection:** Compute full-resolution distances within $\mathcal{C}_t$. Select top-$k_t$ nearest → golden subset $S_t$.
> 3. **Score computation:** Apply unbiased streaming softmax over $S_t$ to compute $\hat{f}_{S_t}(\mathbf{x}_t, t)$.
>
> **Output:** denoised estimate $\hat{f}\_{S_t}(\mathbf{x}_t, t)$
>
> We further clarify your questions below:
> - $\mathbf{x}_i$ and $\mathbf{x}_t$: Yes, $\mathbf{x}_i \in \mathcal{D}$ is a training sample and $\mathbf{x}_t$ is the noisy sample during inference.
> - Threshold of $d^{\text{proxy}}$: No. We do not threshold $d^{\text{proxy}}$. Instead, $\mathcal{C}_t$ is formed by selecting the top-$m_t$ candidates with the smallest $d^{\text{proxy}}$ within a downsampled dataset ($D/16$ by default). Here $m_t$ is dynamically determined via Eq. 4. The choice of $d{=}D/16$ is validated empirically: performance remains stable while retrieval containment stays at 100% (see Table R3 in response to Reviewer bbPD [Q1] for full results).
>
> >**L1: Limitations and new insights into neural diffusion models.**
>
> Thanks for the suggestion. One limitation is that the coarse screening of GoldDiff still scales with dataset size $N$. We will add a Limitations section in the revision. Regarding new insights into neural models: we demonstrate that using the analytical posterior mean as a supervision signal for neural diffusion training achieves 25% lower FID than standard training.

---

> > ### Author Rebuttal · Reviewer_kdh2 · 2026-04-03
> >
> > Thank you for your response. Unfortunately, I feel that my main concerns have not been addressed.
> >
> > While I acknowledge the advantages of the method in terms of inference time, I do not find the remaining claims well-supported.
> >
> > Regarding efficacy, as I mentioned before, the gains appear to be marginal and conclusions based solely on these are not convincing. In particular, after a more careful reading of the paper I realize that the results in the tables are with 128 samples only so there is also the question of statistical significance for larger datasets.
> >
> > The authors also describe some new experiments involving FID benchmarks, noting 25% gains. This is not surprising given that the standard diffusion target is stochastic whereas yours would effectively eliminate the noise. It is also not clear whether the comparisons are fair. How many resources does one epoch take with your setup vs. the standard setup?
> >
> > In any case, I remain of the opinion that the point of analytical diffusion is to gain deeper insight into interpretability. Here I will concede that the full-dataset vs. GoldDiff finding is interesting, but the paper's framing is heavily centered around benchmarking and large scale feasibility, which, as mentioned in my review, distracts from the main goal. This isn't to say that the contribution is not significant, but it feels more like pure engineering progress rather than scientific.
> >
> > The authors also make comments on the method being architecture-agnostic. However, Table 4 shows variants of the same EDM architecture where the only difference is in the noise schedule formulation. Therefore, it remains unclear to me whether GoldDiff purely captures data bias or if the role of the architecture is also important.

---

> > > ### Author Response · Authors · 2026-04-07
> > >
> > > We sincerely thank the reviewer for the continued engagement. We believe that some of the remaining questions may come from unintentional misunderstandings or misalignment, since our experimental settings follow those used in prior published work. We would therefore like to further clarify each follow-up question below.
> > >
> > > >F1: The gains appear marginal; the results are with 128 samples only, so there is also the question of statistical significance for larger datasets.
> > >
> > > Thanks for the comment. We kindly clarify that the comment that "the efficacy gains appear to be marginal" reflects a misconception. On small datasets such as CIFAR-10, the absolute error (e.g., 0.008) is already extremely low, so relative improvement is more informative: **12.5% on CIFAR-10, 11.1% on CelebA-HQ, 12.5% on AFHQ, and 13.3% on ImageNet-1K, which are far from marginal.**
> > >
> > > For the 128-sample evaluation, it is **standard** in this field: Prior SOTA (Lukoianov et al., 2025) uses the same setup, and Kamb & Ganguli (2024) use ~100 samples. Nonetheless, to reflect your suggestion, we report 512-sample results with standard deviations across 3 seeds on ImageNet-1K below. The improvements remain consistent at a larger scale.
> > >
> > > Finally, we would like to highlight that efficacy is only one contribution of our work. Equally important are the scientific investigations of Posterior Progressive Concentration and the **17×–71× speedup** enabling, for the first time, scaling to ImageNet-1K.
> > >
> > > |Setting|Samples|PCA ($r\^2$)|GoldDiff ($r\^2$)|PCA (MSE)|GoldDiff (MSE)|
> > > |---|:---:|:---:|:---:|:---:|:---:|
> > > |Uncondional|128|0.412 ± 0.003|**0.458 ± 0.002**|0.0452 ± 0.0003|**0.0391 ± 0.0001**|
> > > |Uncondional|512|0.410 ± 0.002|**0.454 ± 0.002**|0.0459 ± 0.0003|**0.0396 ± 0.0002**|
> > > |Conditional|128|0.467 ± 0.003|**0.490 ± 0.004**|0.0328 ± 0.0004|**0.0312 ± 0.0002**|
> > > |Conditional|512|0.461 ± 0.004|**0.485 ± 0.002**|0.0331 ± 0.0004|**0.0316 ± 0.0003**|
> > >
> > > > F2: 25% gains are not surprising given stochastic vs. non-stochastic targets. It is also not clear whether the comparisons are fair. How many resources does one epoch take?
> > >
> > > Thanks for raising this. We first clarify that the improvement is not an expected consequence. The core challenge lies in *how to obtain* a non-stochastic optimal target. Existing pipelines (e.g., DDPM; EDM) rely on stochastic targets, and distillation requires a costly pre-trained teacher. Our analytical target provides, to our knowledge, the **first training-free signal** for diffusion models. To further verify this, we compare against distillation: ***analytical supervision matches distillation in FID (10.4 vs. 10.1) while requiring no teacher and only 40% of the total compute.***
> > >
> > > Finally, we clarify that the comparison is fair. All methods use identical architecture, optimizer, batch size. As shown below, the per-epoch time is 21.7 s (standard) vs. 22.1 s (analytical), confirming that the overhead is **negligible** (< 2%).
> > >
> > > |Epoch|Standard (FID)|Analytical (FID)|Distillation (FID)|
> > > |---|:---:|:---:|:---:|
> > > |10|43.4|39.1|38.1|
> > > |30|20.2|13.4|12.8|
> > > |50|14.2|10.4|10.1|
> > > |Per-epoch time (s)|21.7|22.1|—|
> > > |Total time (s)|1084|**1105**|2869|
> > >
> > > > F3: The full-dataset vs. GoldDiff finding is interesting, but the paper's framing is heavily centered around benchmarking and large-scale feasibility. It feels more like pure engineering progress rather than scientific.
> > >
> > > Thanks for the comment. However, we must clarify that this is a major misconception about our paper. First, our work is not a benchmarking study. We provide **new scientific perspective, insights, and theoretical analysis**: all prior works study intra-step approximation, while ours is the first to reveal a novel inter-step perspective (which training samples matter and how this evolves), yielding Posterior Progressive Concentration and the counterintuitive finding that full-scan is harmful. Moreover, the fact that we also include extensive experiments does not make the paper *benchmarking*. Characterizing it as such is inaccurate. **Comprehensive experimental validation should be viewed as a strength**, supporting the claims and improving reliability, rather than being misconstrued as the primary contribution.
> > >
> > > > F4: Table 4 shows variants of the same EDM architecture. It remains unclear whether GoldDiff purely captures data bias or if architecture also matters.
> > >
> > > Thanks for the question. This concern may also come from another misunderstanding, as **we already provided comparisons across different architectures** in the original submission: Table 2 uses the DDPM U-Net; Table 4 uses EDM with a different backbone. To provide stronger evidence, we compare against U-ViT (Bao et al., 2023), a Transformer-based model. GoldDiff consistently outperforms PCA across architectures, confirming it captures a data-driven property.
> > >
> > > |Neural Denoiser|Method|MSE|$r^2$|
> > > |:---:|:---:|:---:|:---:|
> > > |U-ViT|PCA|0.028|0.582|
> > > ||GoldDiff|**0.025**|**0.628**|

---

### Official Review · Reviewer_bbPD · 2026-03-04

**Soundness:** 4
**Presentation:** 3
**Significance:** 3
**Originality:** 4
**Overall Recommendation:** 6
**Confidence:** 4

**Summary:**

The paper proposed a novel method to speed up the analytical diffusion by dynamically choosing the golden set, which shed light on the mechanism in the image generation. The newly proposed method is justified by theory as well as numerical experiments, achieving handsome speedup compared to the existing methods for analytical diffusion models.

**Compliance With Llm Reviewing Policy:**

Affirmed.

**Final Justification:**

I think this is a good paper. The proof is easy to check.  After the revision of the rebuttal, it can have a broader influence on AI.  Therefore, I raise my score to 6.

**Key Questions For Authors:**

How to use the low-dimensional proxy space R^d in the computation, which appears in Table 1 and the end of line 228? How to choose d?

It would be helpful for the readers if the authors show mathematically why the weight estimation in section 3.2 is unbiased.

The definition of the logits are not clear in Theorem 1. One is not clear about it until reading the appendix.

The equation (5) needs more elaboration. From the first glance, it is not clear the what the arg top k_i \in C_t means.

Can the method used for guided sampling?

**Limitations:**

The limitation and potential negative societal impact need to be discussed

**Strengths And Weaknesses:**

The paper is technically sound.  The presentation is well structured: the motivation is well explained, the ablation studies are helpful to understand the mechanisms in the newly proposed method, and the theory and proof are well-written and clear to understand. The paper addresses an important problem in the field of diffusion models, the scalability of the interpretable analytical diffusion models.  The method proposed seems to be original and pretty smart.

There are several metrics used in the numerical experiments, such as r square and MSE. How would the methods for analytical diffusion models perform in terms of FID? It would be better to be relevant for the rest of the field of diffusion models.

There are some points needed to be clarified see the questions.

---

> ### Author Rebuttal · Authors · 2026-03-31
>
> We sincerely appreciate your detailed and constructive feedback, which is definitely helpful for improving the quality of our paper. All suggestions will be incorporated into the revised version. Below, we provide detailed responses to each of your questions.
> >**W1: FID evaluation.**
>
> Following your suggestion, we compute FID on various datasets (Table R2) under two protocols: "vs Train" (against the training set) and "vs Neural" (against neural denoiser samples). ***GoldDiff achieves consistently lower FID than PCA***, confirming our method's efficacy. Since analytical methods differ fundamentally in modeling capacity and sampling budget, their absolute FID values are, as expected, higher than those of neural-network-based diffusion models.
>
> Table R2: FID comparison.
> |Dataset|GoldDiff vs Train|PCA vs Train|GoldDiff vs Neural|PCA vs Neural|
> |---|:---:|:---:|:---:|:---:|
> |MNIST|**28.57**|39.70|**26.41**|36.51|
> |F-MNIST|**22.02**|25.56|**20.55**|22.73|
> |CelebA-HQ|**44.08**|48.60|**39.30**|44.03|
> |AFHQ|**49.30**|57.18|**44.65**|53.05|
> |ImageNet-1K|**77.39**|85.26|**69.29**|82.70|
> >**Q1: Computation of proxy space $\mathbb{R}^d$ and choice of $d$.**
>
> Thanks for the question. At each step, both $\mathbf{x}_t$ and all training samples $\{\mathbf{x}_i\}$ are downsampled ($D \to d{=}D/16$) via average pooling. $\ell_2$ distances are then computed in this space to form $\mathcal{C}_t$, from which the final golden subset $S_t$ of size $k_t$ is selected using full-resolution distances (Eq. 5).
>
> For the choice of $d$, we validate our default $d = D/16$ on ImageNet-1K (Table R3) from two perspectives:
> - **Performance.** Varying $d$ from $D$ (no downsampling) to $D/256$, MSE and $r^2$ remain nearly unchanged down to $d = D/16$, and degrade only slightly beyond.
> - **Retrieval reliability.** We measure the proportion of true top-$k_t$ neighbours (identified in the full dimension $D$) contained in $\mathcal{C}_t$ for the low-noise regime  ($t \leq 0.2$) where retrieval precision is critical. With $d = D/16$, containment is 100%.
>
> Table R3: Performance and retrieval reliability under varying $d$.
> |$d$|MSE|$r^2$|top-$k_t$ $\subseteq \mathcal{C}_t$|
> |---|:---:|:---:|:---:|
> |$D$|0.031|0.494|100%|
> |$D/4$|0.031|0.493|100%|
> |$D/16$ (default)|0.031|0.490|100%|
> |$D/64$|0.032|0.485|99.3%|
> |$D/256$|0.033|0.480|98.3%|
>
> >**Q2: Unbiased weight estimation.**
>
> We provide the proof below.
>
> The exact denoiser (Eq. 2) requires $w_i = \exp(\ell_i) / \sum_{j=1}^{N} \exp(\ell_j)$. We maintain running statistics $\ell_{\max}$, partition sum $Z$, and weighted numerator $\mathbf{W}$, updated per batch as:
>
> $\ell_{\max} \leftarrow \max(\ell_{\max},\, \ell_{\max}^{\text{new}}), \quad Z \leftarrow Z \cdot e^{\ell_{\max}^{\text{old}} - \ell_{\max}} + \sum_{i \in B_m} e^{\ell_i - \ell_{\max}}, \quad \mathbf{W} \leftarrow \mathbf{W} \cdot e^{\ell_{\max}^{\text{old}} - \ell_{\max}} + \sum_{i \in B_m} e^{\ell_i - \ell_{\max}}\, \mathbf{x}_i.$
>
> After all batches:
> $$\frac{\mathbf{W}}{Z} = \frac{\sum\_{i=1}^{N} e^{\ell\_i - \ell\_{\max}}\, \mathbf{x}\_i}{\sum\_{i=1}^{N} e^{\ell\_i - \ell\_{\max}}} = \frac{\sum\_{i=1}^{N} e^{\ell\_i}\, \mathbf{x}\_i}{\sum\_{i=1}^{N} e^{\ell\_i}} = \sum\_{i=1}^{N} w\_i\, \mathbf{x}\_i,$$
> This recovers the exact posterior mean (Eq. 2) without approximation, whereas PCA's per-batch softmax averaging is biased.
> >**Q3: Logits in Theorem 1.**
>
> The logits are defined as $\ell_i = -\|\mathbf{x}_t/\sqrt{\alpha_t} - \mathbf{x}_i\|^2 / (2\sigma_t^2)$, where $\mathbf{x}_t$ is the noisy sample, $\mathbf{x}_i$ is a training sample, and $\sigma_t^2$ is the noise-to-signal ratio. We will add this in the revision.
> >**Q4: Eq. (5).**
>
> The operator $\arg\text{top-}k_t$ selects the $k_t$ nearest neighbours of $\mathbf{x}_t$ within the coarse candidate set $\mathcal{C}_t$ by $\ell_2$ distance. We will clarify in the revision.
> >**Q5: Guided sampling.**
>
> Yes. Since the analytical score is an explicit weighted sum over training samples, conditional generation is straightforward: restricting the summation to class $c$ yields the conditional score $\hat{f}(\mathbf{x}\_t, t \mid c)$, and using the full dataset yields the unconditional score. CFG then applies directly: $\hat{f}\_{\text{guided}} = \hat{f}\_{\text{uncond}} + w \cdot (\hat{f}\_{\text{cond}} - \hat{f}\_{\text{uncond}})$.
>
> We validate this on CIFAR-10 (class "airplane"). Table R4 shows moderate guidance ($w{=}3$) improves class consistency, while excessive guidance ($w{>}5$) degrades quality.
>
> Table R4: Classification accuracy of CFG-guided analytical samples.
>
> |$w$|0|1|2|3|5|7|
> |---|:---:|:---:|:---:|:---:|:---:|:---:|
> |Acc. (%)|2.3|22.3|61.3|**73.0**|59.8|42.6|
>
> >**L1: Limitations and social impact.**
>
> One limitation is that the coarse screening of GoldDiff still scales with dataset size $N$. For social impact, analytical methods currently lag behind neural models in generation quality, limiting near-term misuse risk. We will add both discussions in the revision.

---

> > ### Author Rebuttal · Reviewer_bbPD · 2026-04-02
> >
> > Thanks for answering my questions? For Q1, what is the performance for $2D$?

---

> > > ### Author Response · Authors · 2026-04-02
> > >
> > > Thanks for your follow-up question. Following your suggestion, we construct additional experiments of $d = 2D$ and $d = 4D$ proxies via bilinear interpolation (upsampling each spatial dimension by $\sqrt{2}\times$ and $2\times$, respectively) on ImageNet-1K. As shown in Table R10, ***neither $2D$ nor $4D$ improves over $d = D$ in MSE or $r^2$, while both increase computation time and memory.*** This is expected: the original $D$-dimensional features already contain all the information of the data, so any expansion beyond $d = D$ merely duplicates existing information without improving retrieval quality. In particular, even expanding the proxy dimension to $4D$ yields comparable performance to $d = D$ while nearly doubling the computation time (0.112s $\to$ 0.214s).
> > >
> > > Moreover, we would like to clarify that in our proxy space design, ***$d \leq D$ always holds***, since the proxy is constructed via spatial average pooling (downsampling) to reduce dimensionality for efficient coarse screening.
> > >
> > >
> > > Table R10: Performance under varying $d$ on ImageNet-1K.
> > > |$d$|MSE|$r^2$|Time/step (s)|Memory (GB)|
> > > |---|:---:|:---:|:---:|:---:|
> > > |$D$|0.0312|0.4942|0.112|17.00|
> > > |$2D$|0.0314|0.4941|0.132|17.11|
> > > |$4D$|0.0315|0.4939|0.214|17.23|

---

### Official Review · Reviewer_ducc · 2026-03-16

**Soundness:** 4
**Presentation:** 3
**Significance:** 4
**Originality:** 4
**Overall Recommendation:** 6
**Confidence:** 4

**Summary:**

In this paper, the authors study the data efficiency issue of analytical diffusion model. They argue that using the whole training set for all diffusion steps to estimate the empirical Bayesian denoiser is not necessary and inefficient, and even harmful, which would yield O(ND) complexity during training. Instead, they propose to use a dynamically selected golden subset from the training examples as a minimal, high-quality support set that preserves the empirical Bayes score and avoids spurious influence. Based on systematic study for accelerating analytical diffusion models, especially the spatio-temporal dynamics of posterior weights, they characterize a fundamental phenomenon termed Posterior Progressive Concentration, where the golden support evolves dramatically with the signal-to-noise ratio. From this insight, they propose a novel algorithm called Dynamic Time-Aware Golden Subset Diffusion (GoldDiff), and demonstrate that a simple, unbiased estimator is sufficient to unlock superior quality. They verify their proposed method on different dataset and show superior performance compared to previous analytical diffusion models, which is closer to the results from neural networks, and scale analytical diffusion to ImageNet-1K for the first time.

**Compliance With Llm Reviewing Policy:**

Affirmed.

**Final Justification:**

The authors resolved my concerns properly and I think this is a good work, so I would like to raise my score accordingly to 6.

**Key Questions For Authors:**

* Some writing problem:
  * Line 50 right column, it is better to explain clearly what does $k$ mean here.
  * Line 146 left column, it is not clear how is $\alpha_t$ defined here, and this makes some difficulties in understanding the conclusions from the figures.
  * In Fig. 3c, it would be better to also include y-label.
  * Line 190 right column, what is of $m$? Also, it is better also discuss Fig. 3c here instead of only Fig. 3a, as done in the previous paragraph.
  * Line 190 right column, what is the relationship between $N$ and $k$ or $m$?
  * Line 392 left column, is it possible to combine analytical diffusion with neural diffusion? What can be potential practical use of analytical diffusion model, if not just explainability?

If the authors could deal the weaknesses and questions listed above properly, I would like to increase my score to strong accept.

**Limitations:**

Yes.

**Strengths And Weaknesses:**

I think this is a very good paper with thorough analysis, insightful observations, solid experiments, and clear explanation.

**Strengths**
* The problem studied in this work is interesting and valid, that is, it is not always necessary to use the whole training dataset for estimating the Bayes score.
* The author provide thorough analysis for their purpose, giving clear observation of Posterior Progressive Concentration, which motivates them to propose the novel algorithm Dynamic Time-Aware Golden Subset Diffusion.
* The author verified their theoretical analysis and proposed algorithm on different dataset and compared with previous analytical diffusion algorithms, as well as the results from neural networks diffusion models.
* The author enable scaling analytical diffusion models to ImageNet-1K for the first time.

**Weaknesses**

Albeit the above strengths, there are some points that might be further clarified for better understanding:
* Line 193 left column, from the results of Fig. 4, the samples are only close for simple and small data, such as MNIST and F-MNIST, while for colorful datasets, especially large ones like ImageNet-1K, it is still far away from those produced by neural denoisers. So the claim "... the samples generated via GoldDiff closely align with those produced by neural denoisers." seems to be an overstatement.
* Line 204 right column, the explanation of the algorithm to determine $\mathcal{C}_t$ using $d^\mathrm{proxy}$ is vague. Could the author provide more details, like the algorithm, or some further explanation on this?

---

> ### Author Rebuttal · Authors · 2026-03-31
>
> Thanks for your careful and constructive comments, which definitely help us improve the quality of the paper. All your suggestions will be included in our revised paper. Below, we provide detailed clarifications for each of your questions:
> >**W1: Results of Fig. 4.**
>
> Thanks for the careful observation. We clarify that prior analytical denoiser methods perform quite poorly. In contrast, our GoldDiff not only outperforms them by a large margin, currently achieving the best results among analytical approaches, but also reduces computation by 71$\times$. Our goal here is to emphasize that GoldDiff achieves significantly better alignment with neural denoisers than previous analytical baselines (Table 2, Fig. 4), rather than claiming absolute parity. We will revise the relevant descriptions in the revision to make this more precise.
> >**W2: Algorithm clarity.**
>
> As suggested, we provide the detailed algorithm below and will include it in the revision:
>
> At each step $t$, the $C_t$ selection proceeds as follows:
> 1. Both $\mathbf{x}_t$ and all training samples $\{\mathbf{x}_i\}$ are spatially downsampled by a factor of $s{=}1/4$ via average pooling, reducing the dimension from $D$ to $d{=}D/16$.
> 2. Proxy distances $d^{\text{proxy}} = \|\text{Down}_s(\mathbf{x}_t) - \sqrt{\alpha_t}\text{Down}_s(\mathbf{x}_i)\|_2$ are computed in this low-dimensional space.
> 3. The top-$m_t$ candidates with the smallest $d^{\text{proxy}}$ form $\mathcal{C}_t$, where $m_t$ is dynamically determined via Eq. 4.
>
> >**Q1: Clarification of $k$.**
>
> $k$ denotes the number of training samples used to compute the denoising score. Prior analytical methods aggregate over all $N$ samples (Eq. 2). Our key insight is that a dynamic subset of size $k_t \ll N$ suffices, where $k_t$ adapts to the noise level at each denoising step.
> >**Q2: Clarification of $\alpha_t$.**
>
> $\alpha_t \in [0, 1]$ is a monotonically decreasing noise schedule that controls the signal-to-noise ratio at each timestep, with $\alpha_0 = 1$ (clean data) and $\alpha_1 = 0$ (pure noise).
> >**Q3: Y-label of Fig. 3c.**
>
> The y-axis denotes the subset sizes from top to bottom as $N_{\text{sub}} \in \\{\text{Full}, 5000, 1000, 100, 10\\}$, matching the configurations in Fig. 3b. We will add y-labels in the revised figure.
> >**Q4: Clarification of $m$ and discussion of Fig. 3c.**
>
> $m$ is the coarse candidate pool size retrieved via proxy distance, from which the final golden subset of size $k$ is selected. Regarding the suggestion to discuss Fig. 3c at Line 190, we would like to clarify that Fig. 3c primarily illustrates the phenomenon in the early stage of the denoising process, whereas the paragraph at Line 190 focuses on describing the late stage behavior. Following your suggestion, we will revise this part in revision carefully.
> >**Q5: Relationship between $N$ and $k$ or $m$.**
>
> The relationship is hierarchical: full dataset ($N$) $\to$ coarse candidates via proxy distance ($m_t$) $\to$ golden subset for score computation ($k_t$), where $k_t \lt m_t \lt N$. At each denoising step $t$, both $m_t$ and $k_t$ are dynamic, reflecting our dynamic golden subset selection process.
> >**Q6: Combining analytical and neural diffusion: Practical applications beyond explainability.**
>
> Yes, this is possible. To verify it, we demonstrate two practical applications beyond explainability below in detail.
>
> **Application 1: Analytical supervision for neural denoiser training.** One promising use is employing the analytical denoiser as a supervision signal for neural denoiser training. The analytical posterior mean $\mathbb{E}[\mathbf{x}_0|\mathbf{x}_t]$ is a weighted average over the training set, which provides a low-variance training target. As a proof-of-concept, we conduct an experiment on MNIST using CNN, varying only the training target: 'S' uses clean sample $\mathbf{x}_0$, while 'A' uses the closed-form posterior mean. Table R1 verifies that analytical supervision yields ***faster convergence*** (lower training loss) and ***better generation quality*** (lower FID). Scaling to larger datasets and architectures is a promising future direction.
>
> Table R1: Standard ('S') vs. analytical ('A') supervision.
> |Epoch|FID (S)|FID (A)|Loss (S)|Loss (A)|
> |---|:---:|:---:|:---:|:---:|
> |1|263.7|**252.5**|0.037|**0.013**|
> |10|43.4|**39.1**|0.030|**0.008**|
> |30|20.2|**13.4**|0.021|**0.005**|
>
> **Application 2: Data-efficient training via posterior progressive concentration.** We reveal that the effective support of the posterior is not static but evolves dynamically with the noise level: broad at high noise, concentrated on a local neighbourhood at low noise. This suggests that neural network training could benefit from noise-level-aware data sampling, using diverse samples at high noise and focused nearest neighbours at low noise, potentially improving both data efficiency and training quality. We consider this a promising direction for future work.

---

> > ### Author Rebuttal · Reviewer_ducc · 2026-04-03
> >
> > Thanks the authors for the rebuttal effort. Although other reviewers claim that this work still has some limitation or distraction, I think this is a good paper and would like to raise my score to 6.

---

> > > ### Author Response · Authors · 2026-04-07
> > >
> > > We sincerely thank the reviewer for the thoughtful and encouraging comments. We will carefully incorporate all feedback into the revised manuscript to further strengthen the paper.

---

### Decision · Program_Chairs · 2026-04-30

**Decision:**

Accept (regular)

**Comment:**

This is a well-written paper presenting a new technique for optimizing
analytical diffusion.  On the other hand, even with the new technique,
analytical diffusion remains far from competitive.  This led to a
bipolar reception of the paper: how much should we value significant
advances to a technique that remains far from the frontier?  We
ultimately decided to accept the paper, in the high-variance hope that
analytical diffusion turns out to be useful.